# Protein structure generation via folding diffusion

Kevin E. Wu[1,2,3], Kevin K. Yang[4], Rianne van den Berg[5], Sarah Alamdari[4], James Y. Zou [1,3], Alex X. Lu[4] & Ava P. Amini [4] ✉

The ability to computationally generate novel yet physically foldable protein structures could lead to new biological discoveries and new treatments targeting yet incurable diseases. Despite recent advances in protein structure prediction, directly generating diverse, novel protein structures from neural networks remains difficult. In this work, we present a diffusion-based generative model that generates protein backbone structures via a procedure inspired by the natural folding process. We describe a protein backbone structure as a sequence of angles capturing the relative orientation of the constituent backbone atoms, and generate structures by denoising from a random, unfolded state towards a stable folded structure. Not only does this mirror how proteins natively twist into energetically favorable conformations, the inherent shift and rotational invariance of this representation crucially alleviates the need for more complex equivariant networks. We train a denoising diffusion probabilistic model with a simple transformer backbone and demonstrate that our resulting model unconditionally generates highly realistic protein structures with complexity and structural patterns akin to those of naturally-occurring proteins. As a useful resource, we release an opensource codebase and trained models for protein structure diffusion.

Proteins are critical for life, playing a role in almost every biological process, from relaying signals across neurons[1] to recognizing microscopic invaders and subsequently activating the immune response[2], from producing energy[3] to transporting molecules along cellular highways[4]. Misbehaving proteins, on the other hand, cause some of the most challenging ailments in human healthcare, including Alzheimer's disease, Parkinson's disease, Huntington's disease, and cystic fibrosis[5]. Due to their ability to perform complex functions with high specificity, proteins have been extensively studied as a therapeutic medium[6–8] and constitute a rapidly growing segment of approved therapies[9]. Thus, the ability to computationally generate novel yet physically foldable protein structures could open the door to discovering novel ways to harness cellular pathways and eventually lead to new treatments targeting yet incurable diseases.

Many works have tackled the problem of computationally generating new protein structures, but have generally run into challenges with creating diverse yet realistic folds. Traditional approaches typically apply heuristics to assemble fragments of experimentally profiled proteins into new structures, much like piecing together puzzle pieces[10,11]. Such approaches are limited by the boundaries of expert knowledge and available data. More recently, deep generative models have been proposed for protein structure generation. However, due to the incredibly complex structure of proteins, these commonly do not directly generate protein structures, but rather sets of constraints (such as pairwise distances between residues) that are heavily postprocessed to obtain structures[12,13]. Not only does this add complexity to the design pipeline, but noise in these predicted constraints can also be compounded during post-processing, resulting in unrealistic structures—that is, if the constraints are even satisfiable. Other

[1]Department of Computer Science, Stanford University, Stanford, CA, USA. [2]Center for Personal Dynamic Regulomes, Stanford University, Stanford, CA, USA. [3]Department of Biomedical Data Science, Stanford University School of Medicine, Stanford, CA, USA. [4]Microsoft Research, Cambridge, MA, USA. [5]Microsoft Research, Amsterdam, Netherlands. ✉e-mail: ava.amini@microsoft.com

generative models rely on complex equivariant network architectures or loss functions to learn to generate a 3D point cloud that describes a protein structure[14–19]. Of notable success among these methods is RFDiffusion[18], which not only presents a myriad of conditional generation applications that can design proteins binding specific targets, but also performs thorough experimental validation of computationally generated proteins. Such equivariant architectures can ensure that the probability density from which protein structures are sampled is invariant under translation and rotation. However, translation- and rotation-equivariant architectures are often also symmetric under reflection, leading to violations of fundamental structural properties of proteins like chirality[15]. Intuitively, this point cloud formulation is detached from how proteins biologically fold—by twisting to adopt energetically favorable configurations[20,21].

Here, inspired by the biophysics of the protein folding process, we introduce a generative model that acts on the inter-residue angles in protein backbones instead of on Cartesian atomic coordinates (Fig. 1a, b). This treats each residue as an independent reference frame, thus shifting the equivariance requirements from the neural network to the coordinate system itself. While a similar angular representation has been used in some protein structure prediction works[22–24], it has only received cursory exploration in the context of generative modeling, and only for simple, helix-only protein structures[25]. For generation, we use a denoising diffusion probabilistic model (diffusion model, for brevity)[26,27] with a vanilla transformer parameterization without any equivariance constraints (Fig. 1c). Such models have been highly successful in a wide range of data modalities from images[28,29] to audio[30,31], and are easier to train with better modal coverage than methods like generative adversarial networks (GANs)[32,33]. Combining these ideas, our framework generates backbones by starting from a set of random angles that correspond to a random, unfolded state and iteratively denoising the underlying angles to arrive at a final backbone structure (Fig. 1d). Although this angular denoising procedure does not directly capture any biophysical folding processes, it draws inspiration from how proteins twist and fold into their final structures; as such, we name our method FoldingDiff. We present a suite of validations to demonstrate quantitatively that unconditional sampling from our model directly generates realistic protein backbones—from recapitulating the natural distribution of protein inter-residue angles, to producing overall structures with rich arrangements of multiple structural building block motifs. We show that our generated backbones are diverse and designable, and thus span biologically plausible and interesting protein structures (Fig. 1a). Our work demonstrates the power of biologically inspired problem formulations and represents an important step towards accelerating the development of proteins and protein-based therapies.

## Results

### Representing protein backbones using internal angles

A machine learning method capable of generating protein backbone structures requires a representation to encode structures, as well as a computational model that acts upon that representation. To formulate the FoldingDiff generative model, we propose a simplified framing of protein backbones that inherently embeds geometric invariance within the representation, thus removing the need for complex equivariant networks. We represent a protein backbone structure of $N$ amino acids as a sequence of angle sets comprising 3 bond and 3 dihedral angles formed for each residue, i.e., $x \in [-\pi, \pi)^{(N-1) \times 6}$. That is, each set of six angles describes the relative position of all backbone atoms in the next residue given the position of the current residue's backbone. These six angles are defined precisely in Table 1 and illustrated in Fig. 1b. Notably, these angles do not specify side chain identity or orientation; like other works tackling backbone structure generation, FoldingDiff focuses on designing backbones and relies on external methods to subsequently infer amino acids that fold into designed structures. These internal angles can be easily computed using trigonometry, and converted back to 3D Cartesian coordinates by iteratively adding atoms to the protein backbone[34], fixing bond distances to average lengths (Figure S1). One concern of this iterative reconstruction process is that small errors may accumulate into significant global errors. To rule this out and confirm that our proposed representation can accurately describe longer protein structures, we convert a set of proteins of varying lengths from coordinate to angular representation and back, and find minimal differences between the original and reconstructed coordinates (Figure S2). We similarly investigate the potential for our angular formation to result in structures with atomic clashes, and find that although these clashes do appear, they can be easily

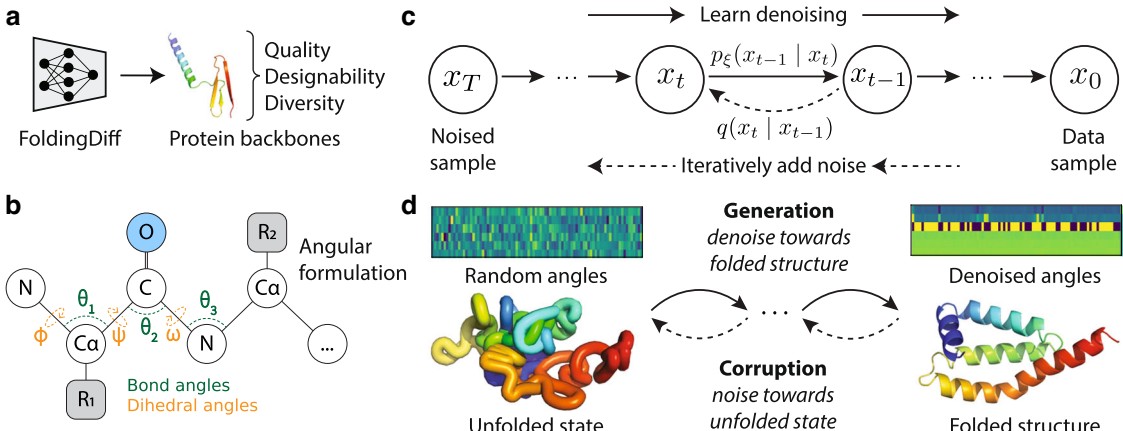

**Fig. 1 | Overview of FoldingDiff. a** FoldingDiff generates protein backbone structures via a diffusion-based generative model. Generated protein backbones are evaluated for their quality, designability, and diversity. **b** Protein backbones are represented as a sequence of bond (green) and dihedral (orange) angle sets. **c** Diffusion models consist of a forward, stochastic noising process and a reverse, learned denoising process. During FoldingDiff's forward process, noise sampled from a wrapped Gaussian (accounting for the periodicity of angles) is iteratively added to an experimentally observed backbone $x_0$ over $T$ discrete steps, where each timestep $t$ adds a small, incremental amount of noise drawn from $q(x_t|x_{t-1})$ until the angles are indistinguishable from a Gaussian wrapped about $[-\pi, \pi)$ at $x_T$. In the reverse process, FoldingDiff is trained to approximate the reverse noise removal procedure $p_\xi(x_{t-1}|x_t)$. **d** Visualization of FoldingDiff's sampling approach. The heatmap on the left represents a set of angles randomly sampled from $[-\pi, \pi)$; these random angles specify a misshapen, unfolded structure represented below the heatmap. FoldingDiff iteratively removes noise from these angles to obtain a structured set of angles depicted in the heatmap on the right, corresponding to the structure in the bottom right colored in rainbow spectrum from N (blue) to C (red) terminus.

**Table 1 | Internal angles used to specify protein backbone structure**

| Angle | Description |
|---|---|
| $\psi$ | Dihedral torsion about $N_i - C\alpha_i - C_i - N_{i+1}$ |
| $\omega$ | Dihedral torsion about $C\alpha_i - C_i - N_{i+1} - C\alpha_{i+1}$ |
| $\phi$ | Dihedral torsion about $C_i - N_{i+1} - C\alpha_{i+1} - C_{i+1}$ |
| $\theta_1$ | Bond angle about $N_i - C\alpha_i - C_i$ |
| $\theta_2$ | Bond angle about $C\alpha_i - C_i - N_{i+1}$ |
| $\theta_3$ | Bond angle about $C_i - N_{i+1} - C\alpha_{i+1}$ |

Some of these involve multiple residues, indicated via $i$ index subscripts. These are illustrated in Fig. 1b.

remedied with common structural relaxation methods (see Supplementary Information; Figure S5, Table S1).

This internal angle formulation has several key advantages. Most importantly, since each residue forms its own independent reference frame, there is no need for an equivariant neural network, as required by diffusion models that work on Cartesian coordinates of protein structure[14,15]. No matter how the protein is rotated or translated, the angles specifying the next atom given the current atom never change. In fact, we demonstrate that our model becomes brittle upon substituting our translation- and rotation-invariant internal angle representation with Cartesian coordinates, keeping all other design choices identical (Figure S6). While extensive data augmentations might help overcome this fragility to some extent, it is simpler and more efficient to use a model intrinsically designed to leverage geometric properties of protein structures.

## Designing and training FoldingDiff

Having defined a simplified but complete angle-based representation of protein structures, our next goal was to train a generative model capable of learning the natural distribution over these sets of angles. We designed a denoising diffusion probabilistic model, or diffusion model for short, capable of generating backbone angles from random noise[26,27]. To learn to do this, diffusion models are trained to iteratively denoise data. During training, starting with a data sample $x_0$, noise is iteratively added over $T$ discrete steps until it is indistinguishable from random noise at $x_T$. In Fig. 1c, this noising procedure is done via the Markov process $q(x_t|x_{t-1})$. The diffusion model is trained to predict the noise added at each step[26], learning a model $p_\xi(x_{t-1}|x_t)$ that performs the reverse denoising process (Fig. 1c). After training is complete, to generate new data points, the diffusion model starts from random noise and applies $T$ steps of iterative denoising where the output of each prior denoising step is used to prepare the input for the next cycle of denoising, culminating in a clean sample (Algorithm 1, Fig. 1d). Importantly, this noising and denoising procedure does not model any biophysical processes of protein folding.

FoldingDiff trains such a diffusion model to generate new protein backbone structures using our angular formulation. Since our diffusion model acts upon periodic angular values, we ensure that noising and denoising procedures are wrapped about the domain $[-\pi, \pi]$. We formulate the denoising model $p$ with a bidirectional transformer model and set $T = 1000$ noising steps. Notably, this transformer architecture does not provide rotation or translation equivariance, as our input representation itself is intrinsically rotation- and translation-invariant. FoldingDiff is trained on a dataset of CATH protein domains[35] between 40 and 128 residues in length; structures with fewer than 40 amino acids are discarded and structures with more than 128 residues are randomly cropped during each training epoch. In total, we were able to successfully train our model using 30,395 unique protein domains, randomly divided into training, validation, and test sets in a 80/10/10 ratio. See Methods for full model specification and training details.

## Generating protein internal angles

After training our FoldingDiff model, we first verified that FoldingDiff generates a realistic distribution of dihedral and bond angles in proteins. We unconditionally generated 10 backbone chains each for every length $l \in [50, 128]$ (see Methods, Fig. 2a, S8), generating a total of 780 backbones. To ensure that the angles generated by our model are general across proteins (and not just memorized from the training dataset), we compared the distributions of angles from these 780 structures to those from a test set of experimental structures not seen during training. To match the length of backbone chains we generated, the test set also consists of structures less than 128 residues in length. We observe that, across all angles, the generated distribution almost exactly recapitulates the test distribution (Fig. 2b, S9). This is true both for angles whose distributions resemble low-variance wrapped Gaussians $(\omega, \theta_1, \theta_2, \theta_3)$ as well as for angles with multi-modal, high-variance distributions $(\phi, \psi)$. Angles with significant mass wrapping about the $-\pi/\pi$ boundary $(\omega)$ are correctly handled as well. Compared to similar plots generated from other protein diffusion methods (see Fig. 3A from Anand and Achim[14]), we qualitatively observe that our method produces a much tighter distribution that more closely matches the natural distribution of bond angles.

However, the individual distributions of each angle alone do not capture the fact that these angles are not independently distributed, but rather exhibit significant correlations. To test if our model correctly captures these correlations, we produced Ramachandran plots of the joint distribution for the the dihedral angles $\phi$ and $\psi$[36]. Figure 2c shows the Ramachandran plot for (experimentally-determined) test set chains with fewer than 128 residues, compared to our 780 generated structures. The Ramachandran plot for natural structures contains three major concentrated regions corresponding to right-handed $\alpha$ helices, left-handed $\alpha$ helices, and $\beta$ sheets. All three of these regions are recapitulated in our generated structures (Fig. 2c), suggesting that FoldingDiff generates all three major secondary structure elements in protein backbones. Furthermore, we see that our model correctly learns that right-handed $\alpha$ helices are much more common than left-handed $\alpha$ helices[37], suggesting that FoldingDiff learns and respects the chirality of protein structures. Prior works that use equivariant networks, such as ref. 15, cannot differentiate between these two types of helices due to network equivariance to reflection.

## Structural characterization of FoldingDiff generations

Our results demonstrate that FoldingDiff generates angles whose individual and joint distributions match those of natural protein structures. However, these prior evaluations only assess if individual pairs of residues form angles consistent with fragments of secondary structures, and not if the overall secondary structure composition of the protein is biologically reasonable, which requires assessing features like the presence and co-occurrence of secondary structures.

We thus sought to assess if the number and co-occurence of secondary structure elements in our generated structures matched those seen in natural backbones. To do this, we used P-SEA[38], a computational algorithm that annotates secondary structure elements for each backbone. We applied P-SEA both to our test set of natural structures and our generated backbones, counted the number of $\alpha$ helices and $\beta$ sheets detected, and measured these secondary structures' frequencies of co-occurrence (Fig. 3a, b). Similar to the natural structures, our generated structures frequently contain multiple secondary structure elements, and exhibit similar co-occurrence patterns to natural structures (e.g., $\alpha$ helices being more common on average compared to $\beta$ sheets). FoldingDiff thus appears to generate rich structural information akin to that of natural protein domains, and does so consistently across multiple independent rounds of generation (Figure S7).

To gain a more nuanced understanding of the types of protein backbone structures generated by FoldingDiff, we visualize the

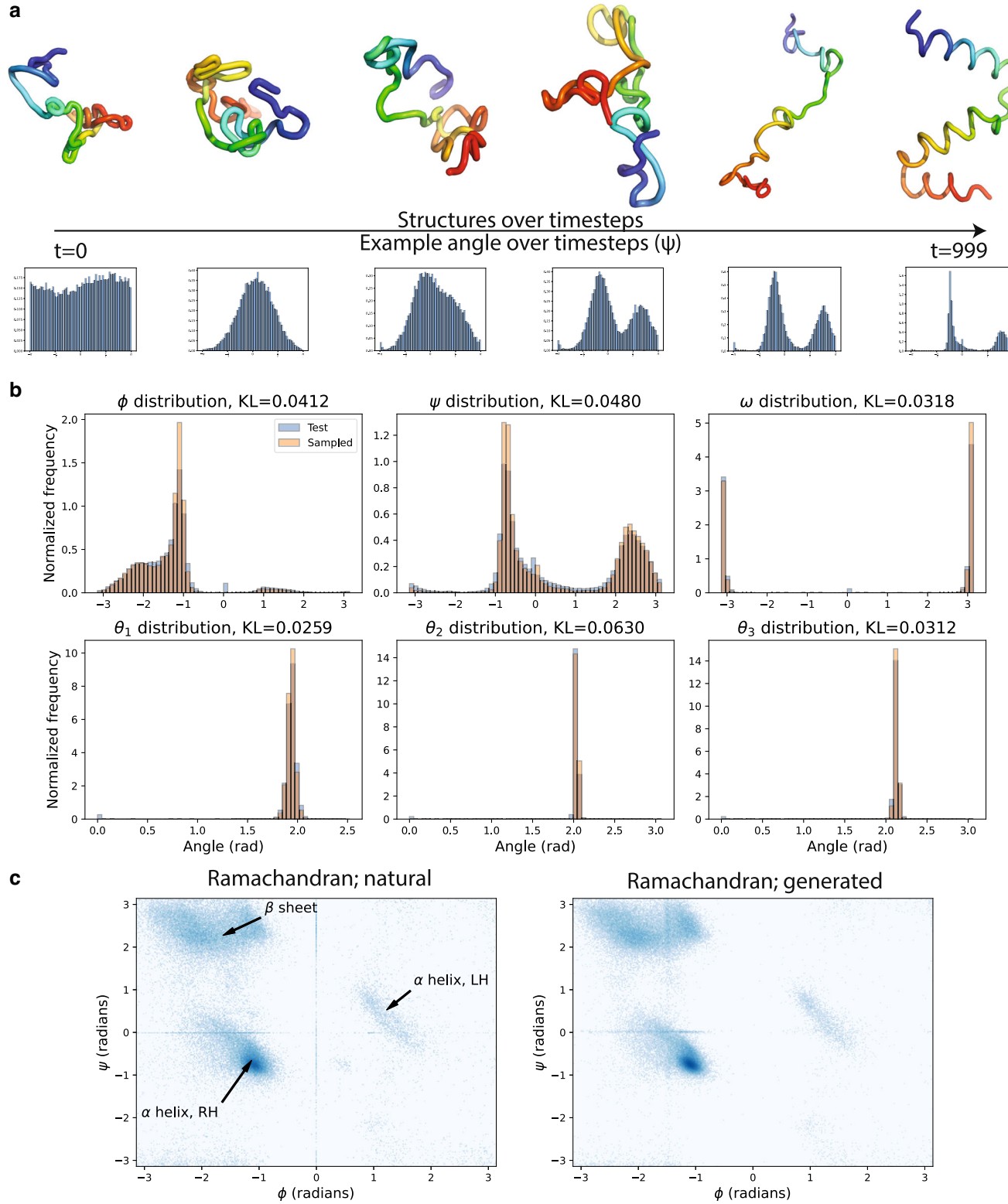

**Fig. 2 | FoldingDiff generates realistic distributions of bond and dihedral angles. a** FoldingDiff iteratively denoises the underlying angles from an unfolded structure (left) towards angles corresponding to a final folded structure (right). Shifts in angle distributions, e.g., of an example dihedral ($\psi$), accordingly occur over this generative process. **b** Distributions of individual angles for backbones sampled from FoldingDiff (Sampled, orange), compared to that of natural backbones (Test, blue). Sampling was repeated 10 times for each backbone length $l \in [50,128]$ yielding a total

of 780 generated backbones. **c** Co-occurrence of $\phi$ and $\psi$ dihedral angles, visualized as Ramachandran plots, over natural proteins (left) and generations sampled from FoldingDiff (right). Arrows indicate ($\phi$,$\psi$) value sets corresponding to three major secondary structure elements: right-handed $\alpha$ helices ($\alpha$ helix, RH), left-handed $\alpha$ helices ($\alpha$ helix, LH), and $\beta$ sheets ($\beta$ sheet). Faint vertical/horizontal lines are artifacts of replacing and imputing null values.

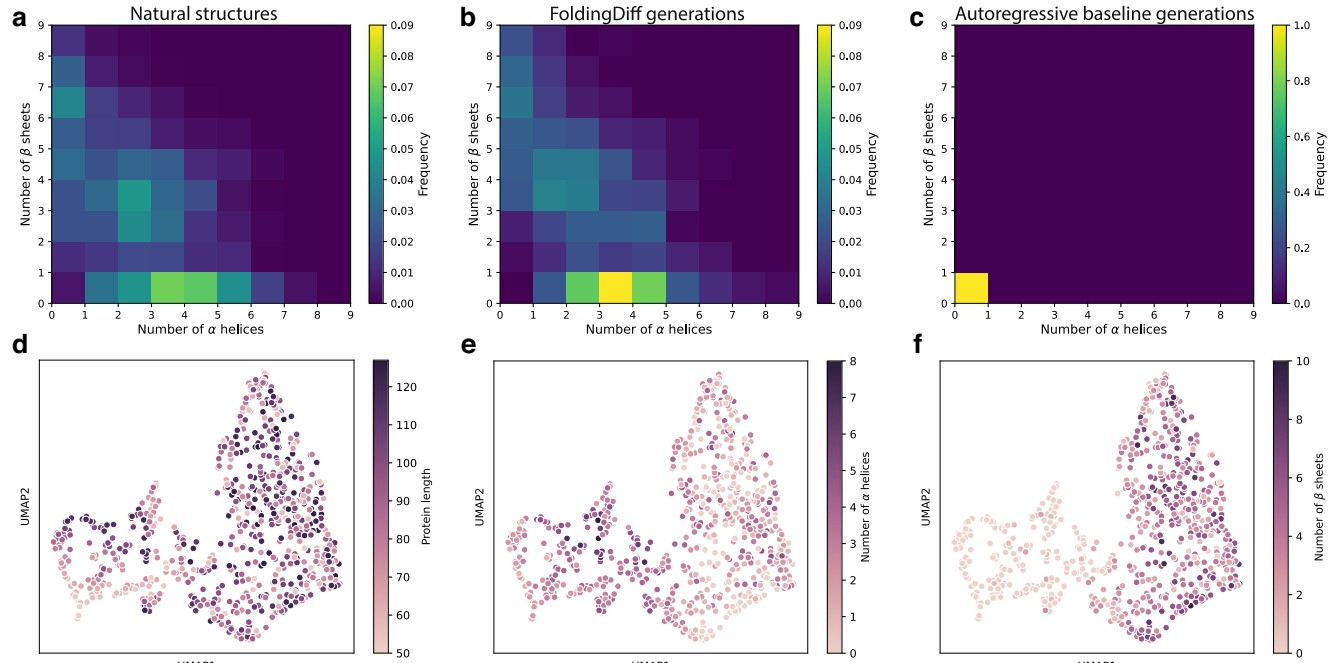

**Fig. 3 | FoldingDiff designs protein backbones with rich secondary structure content. a–c** Secondary structure content within a set of natural protein backbone structures (**a**), generations from FoldingDiff (**b**), and generations from an auto-regressive deep learning model (**c**). The frequency of various combinations of $x$ $\alpha$ helices and $y$ $\beta$ sheets are visualized as a 2D histogram. **d–f** UMAP of Gauss integral embeddings of backbone generations from FoldingDiff ($n = 780$), with individual generations colored by their corresponding backbone length (**d**), number of $\alpha$ helices (**e**), and number of $\beta$ sheets (**f**).

landscape of the generated proteins via their embeddings. Specifically, we embed our generated proteins using 31-dimensional Gauss integrals[39] using the PHAISTOS software suite[40], which we then project to two-dimensions using uniform manifold approximation and projection (UMAP)[41] for visualization. We annotated this plot according to several descriptors—length, number of helices, and number of sheets—and observed that the design space of generated backbones spans a large range of these descriptors (Fig. 3d–f). Jointly visualizing these embeddings in the context of CATH test set structures of similar length (Figure S10) reveals that FoldingDiff's generations share regions of overlap with natural structures while also exploring embedding spaces only sparsely populated by natural structures. This suggests that FoldingDiff has the potential to sample backbones occupying a range of similarities to known structures.

We include several baseline methods to demonstrate that generating backbones with diverse secondary structures is a challenging task. Autoregressive training strategies have been successfully applied to generative modeling of sequences of language tokens in natural language processing[42,43] and amino acid tokens in protein language models[44–46]. Thus, we similarly trained an autoregressive (AR) model using the same angular representation as FoldingDiff (see Methods for additional details) as a baseline generative model over the sequence of angles specifying protein structures. However, annotating the secondary structure content of the AR model's generated structures revealed a failure mode of exclusively producing singular $\alpha$ helices (Fig. 3c, S11, S12). As an additional baseline, we employed a randomized sampling approach, where we shuffle naturally occurring angles and use these shuffled angles to reconstruct a structure (see Methods for additional details). This shuffling preserves the natural distribution of angles and their pairwise correlations (e.g., Ramachandandran plot), but disrupts how they are relatively ordered. This random shuffling baseline produced very few detectable secondary structures; those that are detected are likely a product of random chance (Figure S13). Together, these results demonstrate that FoldingDiff generates protein backbones with secondary structure elements that mirror natural

proteins in their relative frequencies, and that doing so cannot be simply achieved using less principled model designs or naive baselines.

## Designability of generated backbones

Having assessed the biological plausibility of our generated structures from multiple aspects, both at the level of generated angles and the overall secondary structure composition of entire protein backbones, we next sought to assess whether the structures generated by FoldingDiff are designable. In protein design, the designability of a structure reflects whether we can identify, with current methods, an amino acid sequence likely to fold into that designated backbone structure. A generative model that produces a high proportion of designable structures is a more useful model for downstream protein engineering applications.

Previous works have evaluated designability in silico by predicting possible amino acid chains that fold into a generated backbone and evaluating if the structure originating from these sequences matches the original backbone[13,15]. Due to the resource-intensive nature of experimental validations, most works compare generated structures against the structure predicted from sequence by a machine learning model (Fig. 4a)[13,15]. To generate candidate amino acid sequences for a generated structure $s$, we use the ProteinMPNN[47] inverse folding model to output 8 different candidate sequences. For each, we predict the corresponding 3D structure $\hat{s}_1, \ldots, \hat{s}_8$ using the OmegaFold structure prediction method[48]. Finally, we score the structural similarity between the original generated backbone $s$ and predicted structure $\hat{s}$ by computing their TMscore[49], a commonly used metric for evaluating backbone similarity. TMscores range from [0,1] with larger values indicating greater similarity. The maximum score across the 8 candidates $\max_{i \in [1,8]} \text{TMalign}(s, \hat{s}_i)$ is taken as the self-consistency TM (scTM) score. As a TM score $\geq 0.5$ generally indicates the two backbones are in the same protein fold[50], we likewise consider a scTM $\geq 0.5$ to be self-consistent and thus designable (see Methods for additional details).

Using this procedure, we find that 177 of our 780 structures, or 22.7%, are designable with an scTM score $\geq 0.5$ (Fig. 4b) without any

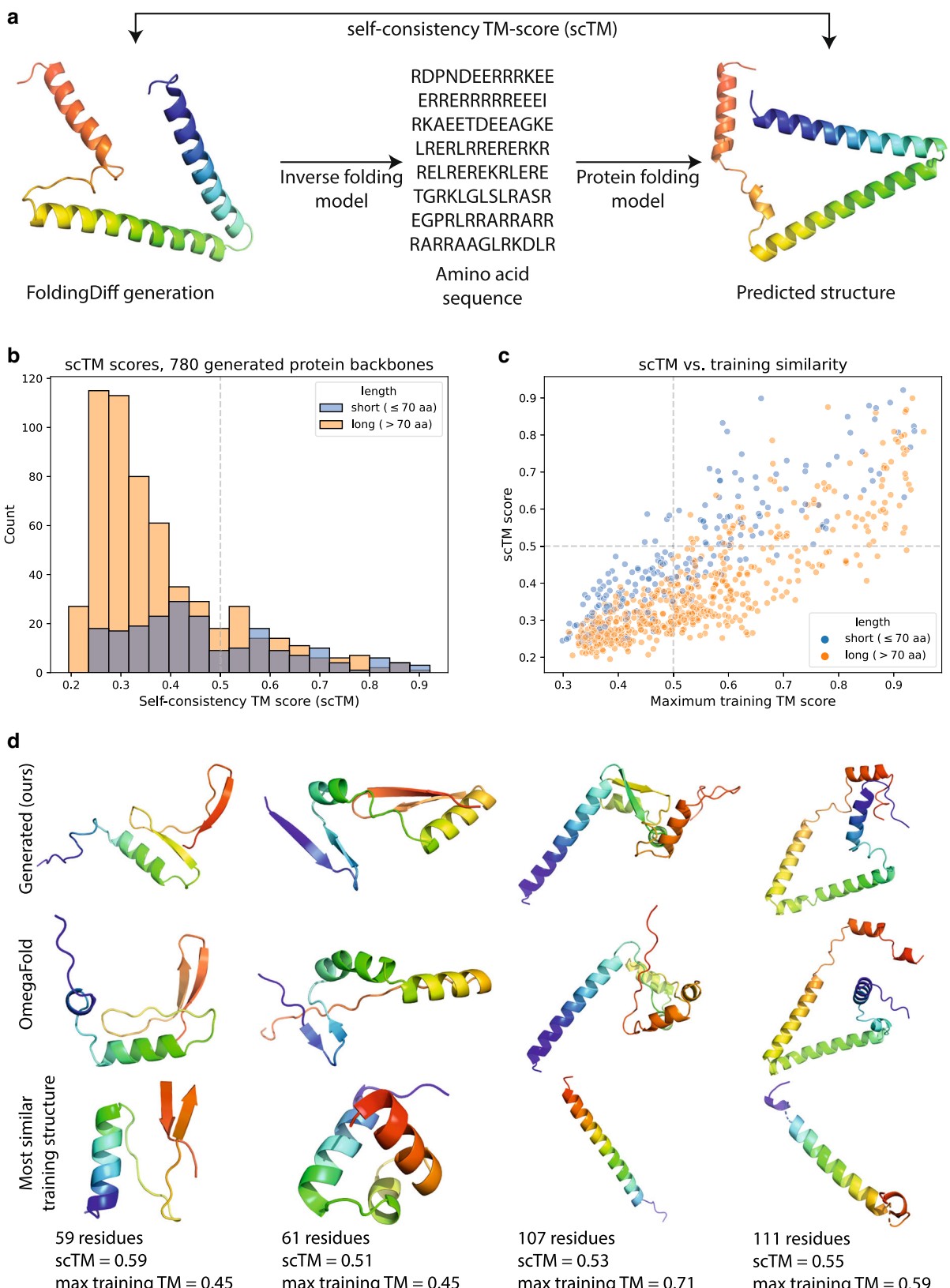

**a** self-consistency TM-score (scTM)

RDPNDEERRRKEE
ERRERRRRREEEI
RKAEETDEEAGKE
LRERLRRERERKR
RELREREKRLERE
TGRKLGLSLRASR
EGPRLRRARRARR
RARRAAGLRKDLR

FoldingDiff generation → Inverse folding model → Amino acid sequence → Protein folding model → Predicted structure

**b** scTM scores, 780 generated protein backbones

**c** scTM vs. training similarity

**d** Generated (ours) / OmegaFold / Most similar training structure

59 residues
scTM = 0.59
max training TM = 0.45

61 residues
scTM = 0.51
max training TM = 0.45

107 residues
scTM = 0.53
max training TM = 0.71

111 residues
scTM = 0.55
max training TM = 0.59

post-processing such as structural refinement or relaxation. This designability is consistent across independent generation runs (Table S2), and is likewise consistent when substituting AlphaFold2 without MSAs[51] in place of OmegaFold (163/780, or 20.9%, designable with AlphaFold2). ProtDiff[15] uses an identical scTM evaluation pipeline leveraging ProteinMPNN and AlphaFold2, and reports a significantly lower proportion of designable structures (92/780 designable, $p = 1.8 \times 10^{-8}$, Chi-square test). Compared to this prior work, FoldingDiff improves the designability of both short sequences (up to 70 residues, 76/210 designable compared to 36/210, $p = 1.7 \times 10^{-5}$, Chi-square test) and long sequences (beyond 70 residues, 87/570 designable compared to 56/570, $p = 5.6 \times 10^{-3}$, Chi-square test). Despite

**Fig. 4 | FoldingDiff generates designable and novel protein backbones.**
**a** Workflow for quantifying backbone designability. The self-consistency (scTM) score is computed as the TMscore similarity between a generated backbone and the predicted structure resulting from an in silico protein design process (inverse folding followed by protein fold prediction). **b** Distribution of scTM scores for all 780 generated backbones, yielding 177 designable backbones with scTM ≥ 0.5, with shorter backbones (≤ 70 aa, blue) exhibiting better designability on average relative to longer backbones (>70 aa, orange). **c** Designability (*y* axis) versus maximum similarity to training set (*x* axis) for 780 backbones generated by FoldingDiff. **d** Representative examples of generated backbones from FoldingDiff (top row), the corresponding closest predicted structure from inverse-folded amino acids (middle row), and the most similar training structure (bottom row). Structures are colored in a rainbow spectrum from N (blue) to C (red) terminus.

these consistent improvements in designability relative to ProtDiff, we similarly find that longer generated structures tend to have poorer designability (Spearman's $\rho = -0.38, p = 3.12 \times 10^{-28}, n = 780$, Figure S14). While ProteinSGM (ESM-IF1 for inverse folding, AlphaFold2 for fold prediction) reports an even higher designability proportion of 50.5%, this value is not directly comparable, as ProteinSGM generates constraints that are subsequently folded using Rosetta[13,52]. Therefore, the designability reported in ProteinSGM does not directly reflect its generative process, as Rosetta post-processing significantly improves the viability of their structures. We also find that confidence scores produced by OmegaFold and AlphaFold2 correlate well with scTM (Figure S16), suggesting that applications leveraging FoldingDiff could directly use these scores to identify high-quality generations.

To further contextualize the designability ratio achieved by FoldingDiff, we similarly evaluate scTM scores for the set of structures previously obtained using randomized angle shuffling. None of these randomized structures are designable, and the scTM scores are significantly lower than those produced by FoldingDiff ($p = 1.6 \times 10^{-121}$, two-sided Mann–Whitney test, $n = 1560$, Figure S17). Conversely, we evaluate experimental structures to establish an upper bound for designability. 87% of natural structures have an scTM ≥ 0.5 (Figure S17). The fact that even real structures do not have perfect designability demonstrates that the scTM evaluation pipeline can be fragile, and that some fraction of FoldingDiff's generations may produce interesting, foldable proteins even if the scTM pipeline does not identify them as such. Finally, we evaluate the designability of structures produced by the aforementioned autoregressive baseline. While 89% of these structures have an scTM ≥ 0.5 (Figure S12b), these autoregressive generations are entirely comprised of singular $\alpha$ helices (Figures S11, S12a), making these structures biologically uninteresting even if they might be designable. Finally, we find that of our 177 backbones considered to be designable with OmegaFold, only 16 contain $\beta$ sheets as annotated by P-SEA. Conversely, of the remaining 603 backbones with scTM < 0.5, 533 contain $\beta$ sheets. This suggests that FoldingDiff may have greater difficulty generating high scTM structures with $\beta$ sheets ($p = 4.6 \times 10^{-91}$, Chi-square test, Figure S15). Although we believe that some degree of this is due to the fragility of the scTM pipeline itself, there is certainly much work that can be done to improve FoldingDiff's ability to model complex arrangements of beta sheets.

### Novelty and diversity of protein backbones generated by FoldingDiff

The observed modal collapse of the autoregressive baseline strongly illustrates the importance of measuring the diversity of generated structures. To this end, we evaluate FoldingDiff's generative diversity using two metrics: novelty with respect to the training set of natural backbones and diversity spanned by the pool of generated backbones. To measure similarity to training data, we calculate the maximum TM score of each generated backbone to any training set structure. We observe that the maximum training TM score is significantly correlated with scTM (Spearman's $\rho = 0.78$, $p = 7.9 \times 10^{-165}$, Fig. 4c), indicating that structures more similar to the training set tend to be more designable. This correlation is in part consistent with the observation that the scTM pipeline can be fragile and may fail even for natural structures (Figure S17)—if our generated structures deviate at all from the

data distribution of native proteins, as is the case for generations with low training TM scores, such failures are invariably even more likely. Moreover, the distribution of training TM scores indicates that FoldingDiff does not merely memorize the training structures; doing so would result in a distribution of training TM scores much more heavily skewed towards 1.0, as is reported by methods such as ProteinSGM[13].

Having evaluated designability and novelty through TM scores (Figs. 4b, c), we next qualitatively explored the relationship between the designability and novelty of generated protein backbone structures (Fig. 4d). Among the three rows of representative generated structures illustrated, the top row shows 4 distinct designable backbones generated by FoldingDiff; for each of these and their corresponding columns, the middle row shows the best-matching OmegaFold predictions from the scTM pipeline, and the bottom row shows the most similar training structure by TM score. Comparing these structures suggests that FoldingDiff's generations may not be as similar to the training set as training TM scores indicate. For example, considering the second structure, FoldingDiff's generation contains two pairs of antiparallel $\beta$ sheets not present in the closest training structure; similarly, for the third structure, our generated structure contains several alpha helices, whereas the closest training structure contains only a single helix. In addition, OmegaFold's predicted folds (middle row) are consistently very similar to FoldingDiff's initial generated backbone in each of these cases, qualitatively suggesting that designability also may be greater than scTM scores themselves might suggest. Figure S18 additionally shows randomly selected structures spanning the entire range of scTM designability scores. Overall, we observe that structures with very poor designability (scTM <0.25) tend to include long, unstructured loop regions connecting interspersed regions of beta sheets. Structures with high designability (scTM ≥ 0.75) are not very diverse and tend to incorporate mostly alpha helices with minimal, if any, kinks and turns. Structures in the middle of this range (0.25 ≤ scTM <0.75) appear qualitatively reasonable and contain a strong variety of combinations of secondary structure motifs.

To quantitatively measure the structural diversity spanned by FoldingDiff's generations, we cluster all generated designable backbones (scTM ≥ 0.5) according to their pairwise TM scores (see Methods for additional details). The resulting clustering and pairwise distance heatmap (Figure S19a) suggests that FoldingDiff's designable backbones do not typically share large degrees of structural similarity, and are thus structurally diverse. In fact, when compared to a similar clustering of naturally occurring protein structures (Figure S19b), FoldingDiff achieves a similar level of diversity. This sharply contrasts with prior works[15], whose generated protein structures appear to mainly consist of minor variations on a handful of core structural motifs, and with our autoregressive baseline, which exhibits extremely poor diversity (Figure S19c). Overall, these results demonstrate that FoldingDiff generates high-quality backbones that are designable, diverse, and may include structures that are meaningfully different from its training set. These three properties are hallmarks of a strong generative model that can effectively explore the space of proteins outside what is already known from biology.

### Discussion
In this work, we present a parameterization of protein backbone structures that we couple with a powerful generative deep learning model to

enable effective sampling of novel, diverse, and realistic protein backbone structures. Considering each residue to be its own reference frame, we describe a protein using the resulting relative internal angle representation. We show that a standard transformer can then be used to build a diffusion model that generates high-quality, biologically plausible, and diverse protein structures. These generated backbones respect protein chirality and exhibit high designability.

While we demonstrate promising results with our model, we note limitations to our method that motivate opportunities for future work. Although we show empirically that our angular representation can be trained to generate high-quality backbones with up to 128 residues, there is no guarantee that our framework scales effectively to longer backbone lengths or more complex applications, such as modeling much larger proteins with intricate arrangements of secondary structures like $\beta$ barrels. Namely, although formulating a protein as a series of angles enables use of simpler models without equivariance mechanisms, this framing allows for single-angle errors to significantly alter the overall generated structure—a sort of lever arm effect (see Supplementary Information). Additionally, some generated structures exhibit collisions where the generated structure crosses through itself (see Supplementary Information). Future work could explore methods to avoid these pitfalls using geometrically-informed architectures such as those used in[48]. Our generated structures are still of relatively short lengths compared to natural proteins, which typically have several hundred residues; future work could extend towards longer structures, potentially incorporating additional losses or inputs that help checkpoint the structure and reduce accumulation of error. Future work could also build upon FoldingDiff's backbone generation functionality to additionally perform amino acid sequence and side-chain conformation generation, rather than relying on external inverse folding methods.

Further work that guides or biases the generative process towards backbones with desirable protein-protein interactions, structural domains, or functional traits will help realize the potential of proteins as therapeutic agents. Similarly, extending FoldingDiff to perform structure diffusion guided by amino acid sequence could enable new advances in protein fold prediction. Despite their lauded success, models like AlphaFold2 have been shown to perform poorly on proteins with highly dynamic folds, such as intrinsically disordered proteins[53,54]. The conventional method for computationally modeling these systems involves extensive molecular dynamics simulations that are orders of magnitude slower than a typical deep learning model. A sequence-guided generative model of structure might bypass such expensive simulations and directly sample from the conditional distribution of conformations matching a given amino acid sequence. Indeed, some works have already begun to leverage FoldingDiff's ability to rapidly generate many backbones to model disordered protein structural ensembles[55].

In summary, our work provides an important step in leveraging biologically inspired problem formulations for generative protein design. Future work to develop and apply FoldingDiff and related methods will unlock new capabilities not only in accelerating therapeutic development, but also in rapidly exploring the structural space of proteins, allowing for advances in fields like molecular dynamics. More broadly, we envision that our method provides a framework for how biologically grounded deep learning methods can lead to effective, powerful solutions to outstanding biomedical challenges.

## Methods
### Denoising diffusion probabilistic models
Denoising diffusion probabilistic models (or diffusion models, for short) leverage a Markov process $q(x_t|x_{t-1})$ to corrupt a data sample $x_0$ over $T$ discrete timesteps until it is indistinguishable from noise at $x_T$. A diffusion model $p_\xi(x_{t-1}|x_t)$ parameterized by $\xi$ is trained to reverse this forward noising process, denoising pure noise towards

samples that appear drawn from the native data distribution[27]. Diffusion models were first shown to achieve good generative performance by ref. 26; we adapt this framework for generating protein backbones, introducing necessary modifications to work with periodic angular values.

We modify the standard Markov forward noising process that adds noise at each discrete timestep $t$ to sample from a wrapped normal instead of a standard normal[56]:

$$q(x_t|x_{t-1}) = \mathcal{N}_{\text{wrapped}}\left(x_t; \sqrt{1-\beta_t}x_{t-1}, \beta_t I\right)$$
$$\propto \sum_{k=-\infty}^{\infty} \exp\left(\frac{-||x_t - \sqrt{1-\beta_t}x_{t-1} + 2\pi k||^2}{2\beta_t^2}\right)$$

where $\beta_t \in (0,1)_{t=1}^T$ are set by a variance schedule. We use the cosine variance schedule[33] with $T = 1000$ timesteps:

$$\beta_t = \text{clip}\left(1 - \frac{\bar{\alpha}_t}{\bar{\alpha}_{t-1}}, 0.999\right) \bar{\alpha}_t = \frac{f(t)}{f(0)} f(t) = \cos\left(\frac{t/T+s}{1+s} \cdot \frac{\pi}{2}\right)$$

where $s = 8 \times 10^{-3}$ is a small constant for numerical stability. We train our model for $p_\xi(x_{t-1}|x_t)$ with the simplified loss proposed by ref. 26, using a neural network $\text{nn}_\xi(x_t,t)$ that predicts the noise $\epsilon \sim \mathcal{N}(0,I)$ present at a given timestep (rather than the denoised mean values themselves). To handle the periodic nature of angular values, we introduce a function to wrap values within the range $[-\pi,\pi)$: $w(x) = ((x+\pi)\text{mod}2\pi) - \pi$. We use $w$ to wrap a smooth L1 loss[57] $L_w$, which behaves like L1 loss when error is high, and like an L2 loss when error is low; we set the transition between these two regimes at $\beta_L = 0.1\pi$. While this loss is not as well-motivated as torsional losses proposed by ref. 56, we find that it achieves strong empirical results.

$$d_w = w\left(\epsilon - \text{nn}_\xi\left(w\left(\sqrt{\bar{\alpha}_t}x_0 + \sqrt{1-\bar{\alpha}_t}\epsilon\right),t\right)\right)$$
$$L_w = \begin{cases} 0.5\frac{d_w^2}{\beta_L} & \text{if}|d_w|<\beta_L \\ |d_w| - 0.5\beta_L & \text{otherwise} \end{cases}$$

During training, timesteps are sampled uniformly $t \sim U(0,T)$. We normalize all angles in the training set to be zero mean by subtracting their element-wise angular mean $\mu$; validation and test sets are shifted by this same offset.

Figure 1 illustrates this overall training process, including our previously described internal angle framing. The internal angles describing the folded chain $x_0$ are corrupted until they become indistinguishable from random angles, which results in a disordered mass of residues at $x_T$; we sample points along this diffusion process to train our model $\text{nn}_\xi$. Once trained, the reverse process of sampling from $p_\xi$ also requires modifications to account for the periodic nature of angles, as described in Algorithm 1. The variance of this reverse process is given by $\sigma_t = \sqrt{\frac{1-\bar{\alpha}_{t-1}}{1-\bar{\alpha}_t} \cdot \beta_t}$.

**Algorithm 1.** Sampling from $p_\xi$ with FoldingDiff
1: $x_T \sim w(\mathcal{N}(0,I))$ ▷ Sample from a wrapped Gaussian
2: **for** $t = T,...,1$ **do**
3: $z = \mathcal{N}(0,I)$ if $t > 1$ else $z = 0$
4: $x_{t-1} = w\left(\frac{1}{\sqrt{\alpha_t}}\left(x_t - \frac{1-\alpha_t}{\sqrt{1-\bar{\alpha}_t}}\text{nn}_\xi(x_t,t)\right) + \sigma_t z\right)$ ▷ Wrap sampled values about[-π, π]
5: **end for**
6: **return** $w(x_0 + \mu)$ ▷ Un-shift generated values by original mean shift.

This sampling process can be intuitively described as refining internal angles from an unfolded state towards a folded state.

## Modeling and dataset

For our reverse (denoising) model $p_\xi(x_t, t)$, we adopt a vanilla bidirectional transformer architecture[58] with relative positional embeddings[59]. Our six-dimensional input is linearly upscaled to the model's embedding dimension ($d = 384$). To incorporate the timestep $t$, we generate random Fourier feature embeddings[60] as done in ref. [61] and add these embeddings to each upscaled input. To convert the transformer's final per-position representations to our six outputs, we apply a regression head consisting of a densely connected layer, followed by GELU activation[62], layer normalization, and finally a fully connected layer outputting our six values. In total, our model has 14.56 million trainable parameters. As is typical of transformer architectures, we use attention masking to allow our model to batch across inputs of variable lengths (both during training and generation). We train this network with the AdamW optimizer[63] over 10,000 epochs, with a learning rate that linearly scales from 0 to $5 \times 10^{-5}$ over 1000 epochs, and back to 0 over the final 9000 epochs. Validation loss appears to plateau after $\approx 1400$ epochs; additional training does not improve validation loss, but appears to lead to a poorer diversity of generated structures. We thus take a model checkpoint at 1488 epochs for all subsequent analyses.

We train our model on the CATH dataset, which provides a de-duplicated set of protein structural folds spanning a wide range of functions where no two chains share more than 40% sequence identity over 60% overlap[35]. We exclude any chains with fewer than 40 residues. Chains longer than 128 residues are randomly cropped to a 128-residue window at each epoch. A random 80/10/10 training/validation/test split yields 24,316 training backbones, 3039 validation backbones, and 3040 test backbones. Note that since we do not use test set accuracy or reconstruction as a primary metric for evaluating our work, potential overlaps and similarities between training and test data does not artificially inflate any of the results we report.

## Designability evaluation and self-consistency TM score

Our self-consistency TM score (scTM) evaluation pipeline is similar to previous evaluations done by ref. [15] and [13], with the primary difference that we use OmegaFold[48] instead of AlphaFold[51]. OmegaFold is designed without reliance on multiple sequence alignments (MSAs), and performs similarly to AlphaFold while generalizing better to orphan proteins that may not have such evolutionary neighbors[48]. Furthermore, given that prior works use AlphaFold without MSA information in their evaluation pipelines, OmegaFold appears to be a more appropriate method for scTM evaluation.

OmegaFold is run using default parameters (and 'release1' weights). We also run AlphaFold without MSA input for benchmarking against[15]. We provide a single sequence reformatted to mimic a MSA to the colabfold tool[64] with 15 recycling iterations. While the full AlphFold model runs 5 models and picks the best prediction, we use a singular model (model1) to reduce runtime.

Ref. [15] use ProteinMPNN[47] for inverse folding and generate 8 candidate sequences per structure, whereas ref. [13] use ESM-IF1[65] and generate 10 candidate sequences for each structure. We performed self-consistency TM score evaluation for both these methods, generating 8 candidate sequences using author-recommended temperature values ($T = 1.0$ for ESM-IF1, $T = 0.1$ for ProteinMPNN). We use OmegaFold to fold all amino acid sequences for this comparison. We found that ProteinMPNN in $C_\alpha$ mode (i.e., alpha-carbon mode) consistently yields much stronger scTM values (Tables S2, S3); we thus adopt ProteinMPNN for our primary results. While generating more candidate sequences leads to a higher scTM score (as there are more chances to encounter a successfully folded sequence), we conservatively choose to run 8 samples to be directly comparable to ref. [15]. We also use the same generation strategy as ref. [15], generating 10 structures for each structure length $l \in [50, 128]$ —thus the only difference in our scTM analyses is the generated structures themselves.

## Shuffling angles to generate "random" structures

To contextualize FoldingDiff's generations, we implement a naive angle generation baseline. We take our test dataset angles, and concatenate all examples into a matrix of $\hat{x} \in [-\pi, \pi]^{\hat{N} \times 6}$, where $\hat{N}$ denotes the total number of angle sets in our test dataset, aggregating across all individual chains. To generate a backbone structure of length $l$, we simply sample $l$ indices from $U(0, \hat{N})$. This creates a chain that perfectly matches the natural distribution of protein internal angles, while also perfectly reproducing the pairwise correlations, i.e., of dihedrals in a Ramachandran plot, but critically loses the correct ordering of these angles. We randomly generate 780 such structures (10 samples for each integer value of $l \in [50, 128]$). This is the same distribution of lengths as the generated set in our main analysis. For each of these, we perform secondary structure annotation as well as scTM evaluation.

## Autoregressive baseline model

As a baseline method for our generative diffusion model, we also implemented an autoregressive (AR) transformer $f_{AR}$ that predicts the next set of six angles in a backbone structure (i.e., the same angles used by FoldingDiff, described in Fig. 1b and Table 1) given all prior angles.

Architecturally, this model consists of the same transformer backbone as used in FoldingDiff combined with the same regression head converting per-token embeddings to angle outputs, though it is trained using absolute positional embeddings rather than relative embeddings as this improved validation loss. The total length of the sequence is encoded using random Fourier feature embeddings, similarly to how time was encoded in FoldingDiff, and this embedding is similarly added to each position in the sequence of angles. The model, which consists of 14.41 million trainable parameters, is trained to predict the $i$-th set of six angles given all prior angles, using attention masking to hide the $i$-th angle and onwards. We use the same wrapped smooth L1 loss as our main FoldingDiff model to handle the fact that these angle predictions exist in the range $[-\pi, \pi)$; specifically: $L_w(x^{(i)}, f_{AR}(x^{(0,...,i-1)}))$ where superscripts indicate positional indexing. This approach is conceptually similar to causal language modeling[66], with the important difference that the inputs and outputs are continuous values, rather than (probabilities over) discrete tokens.

This model is trained using the same dataset and data splits as our main FoldingDiff model with the same preprocessing and normalization. We train $f_{AR}$ using the AdamW optimizer with weight decay set to 0.01. We use a batch size of 256 over 10,000 epochs, linearly scaling the learning rate from 0 to $5 \times 10^{-5}$ over the first 1000 epochs, and back to 0 over the remaining 9000 epochs. We find that the validation loss does not improve beyond 1446 epochs of training, and use this model checkpoint for generation.

To generate structures from $f_{AR}$, we seed the autoregressive model with 4 sets of 6 angles taken from the corresponding first 4 angle sets in a randomly chosen naturally occurring protein structure. This serves as a random, but biologically realistic, prompt for the model to begin generation. We then supply a fixed length $l$ and repeatedly run $f_{AR}$ to obtain the next $i$-th set of angles, appending each prediction to the existing $i - 1$ values in order to predict the $i + 1$ set of angles. We repeat this until we reach our desired structure length.

## Clustering protein backbones

To evaluate the diversity of a set of protein backbones, we cluster them according to a pairwise distance metric between proteins $p_1, p_2$ defined as $d = 1 - \text{TMscore}(p_1, p_2)$. After calculating the pairwise distance matrix between all proteins in the set of backbones, we apply hierarchical clustering with average linkage. This clustering is the same as performed by ref. [15], which allows us to directly compare clustering results and plots. We apply this clustering procedure to protein sets

generated by FoldingDiff, as well as a comparable set of naturally occurring proteins to establish a reference.

### 3D visualization of protein structures

3D visualizations of protein structures are done via PyMOL[67]. For CATH structures and structures generated by AlphaFold2 and OmegaFold, secondary structure cartoons are drawn based on annotations from PyMOL's built-in "dss" method. Structures generated by FoldingDiff do not have side chain information, and thus do not contain the hydrogen bonding information required for "dss" to work properly. To illustrate FoldingDiff's final generations, we instead draw secondary structures as annotated by P-SEA[38], which we also use throughout our manuscript for secondary structure evaluation. We additionally note that oxygen atoms must be present for PyMOL's cartoons to display properly; thus, we insert oxygen atoms in our generated backbone structures (which canonically include only $N - C\alpha - C$ atoms) in a coplanar, trans configuration with respect to the peptide bond[68].

### Reporting summary

Further information on research design is available in the Nature Portfolio Reporting Summary linked to this article.

## Data availability

The data that support this study are available from the corresponding authors upon request. The CATH dataset (version 4.3.0) used in this study to train the FoldingDiff model is publicly available [http://download.cathdb.info/cath/releases/all-releases/v4_3_0/non-redundant-data-sets/], and a copy is deposited in Zenodo at https://doi.org/10.5281/zenodo.8388270 [https://zenodo.org/records/8388270]. The 780 structures generated in this study along with metadata tables with designability scores, training set similarities, secondary structure content, and Gauss integral embeddings (for both FoldingDiff generations and test set structures) are deposited in Zenodo at https://doi.org/10.5281/zenodo.8388286 [https://zenodo.org/records/8388286]. We also reference the PDB structure 1CRN (https://doi.org/10.2210/pdb1CRN/pdb).

## Code availability

All code for training FoldingDiff and performing downstream analyses is implemented in Python (version 3.8) and various open-source packages, notably PyTorch (version 1.12)[69] and PyTorch Lightning (version 1.6.4)[70] for modeling, and biotite (version 0.34)[71], scikit-learn (version 1.2.1)[72], numpy (version 1.22.3)[73], and pandas (version 1.1.5)[74,75] for analysis. Plots were generated using matplotlib[76], seaborn[77], and PyMOL (version 2.5.4)[67]. All training and sampling code, trained model weights, plotting code[78–81], and scripts to generate all results in this manuscript are available at [https://github.com/microsoft/foldingdiff] and are citeable on Zenodo at https://doi.org/10.5281/zenodo.10365890 [https://zenodo.org/records/10365890].

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

## Acknowledgements

This work was primarily the outcome of an internship by K.E.W. at Microsoft Research. Models were trained and evaluated using computational resources generously provided by Microsoft and Stanford University. K.E.W. and J.Y.Z. were supported by the Chan Zuckerberg Investigator Award. We thank Nicolo Fusi for valuable feedback and insight.

## Author contributions

K.E.W., K.K.Y., A.X.L. and A.P.A. initiated, conceived, and designed the work. K.E.W. performed modeling and analyses, with input from all authors. K.K.Y., R.vdB., S.A. and A.P.A. provided guidance on diffusion models and their implementation. K.K.Y., S.A., J.Z., A.X.L. and A.P.A. provided guidance on evaluation methods. K.K.Y., A.X.L. and A.P.A. supervised the research. K.E.W., K.K.Y. and A.P.A. wrote the first draft of the manuscript. All authors contributed to writing and editing subsequent drafts of the manuscript and approved the final manuscript. Work by K.E.W. was done principally during an internship at Microsoft Research.

## Competing interests

The authors declare no competing interests.
