## [Peer Review File · Nature Communications]

REVIEWER COMMENTS

Reviewer #1 (Remarks to the Author):

The authors propose a diffusion model named FoldingDiff for protein structure generation. FoldingDiff is a standard denoising diffusion probabilistic model that uses a generic transformer to model the backbone torsion and bond angles. They show that this can be used to generate protein backbones, about 22% of which are designable using inverse folding models such as ProteinMPNN.

Progress in the field of ML-based protein structure modeling/design is extremely rapid now (also see below), and it is a highly exciting time (and the authors are certainly among those contributing to the progress and excitement); this also makes reviewing (as well as authoring) papers a bit more difficult. FoldingDiff was one of the earliest protein backbone diffusion models out in preprint, and the authors took their own, unique approach to the problem by focussing on a backbone description with angles, with its own advantages and disadvantages from which the field can learn.

For this reason, I would be in favor of publication if the points below can be addressed.

Major comments:

- While FoldingDiff was out in preprint well ahead of models such as RFDiffusion, Chroma, Genie or FrameDiff, I do think it would be proper to cite and discuss at least RFDiff which is out now as a journal publication (I realize with the speed of current progress, it may be difficult to keep up with all the preprints).
- Since torsion/bond angles are invariant representations that only implicitly capture its 3D localization, I am curious how well the model learns to prevent atomic clashes compared to point cloud or AF2 frame-based models. Please provide some quantification of atomic clashes (ex. % of atoms < literature atomic radii) and comparisons to other models.
- On the issue of error accumulation (line 91-96): I don't think this is the proper way to test whether the lever effect is not an issue, since all this is doing is going from native 3D coordinates to internal coordinates and back. If some error is introduced by a model and the internal coordinates are imperfect, it will likely affect the downstream coordinates significantly in terms of RMSD/TM. Thus, it would be interesting to see the reconstruction error when the angles are perturbed by increasing levels of Gaussian noise, and observe how robust this representation is to noise.

Minor comments:

- Since FoldingDiff predicts the omega torsion angle which in turn learn the correct placement of oxygen atoms, I am not sure why pyMOL fails to annotate secondary structures in the protein structures, which should default to proper secondary structure visualization given correct hydrogen bonding.
- Backbone diversity visualization (line 175-181): A UMAP projection solely of generated samples seems less informative – it would be interesting to add native structures to the projections and analyze the differences in coverage of the various descriptors.

- Baseline (line 190-195): I am honestly not sure how valid this baseline is – it seems pretty obvious that this baseline will not generate proper protein structures even if the angles may be sampled from the proper angular distribution. The autoregressive baseline seems sufficient.
- Designability (line 254-256): One concern I have is that a lower distribution of training TM-scores may be due to FoldingDiff not generating unrealizable protein structures, which can be argued given the ~22% scTM scores. Do structures with scTM < 0.5 at least visually resemble realistic structures? It would also be nice to see some randomly selected samples at various scTMs in the supplementary.

Reviewer #2 (Remarks to the Author):

In “Protein structure generation via folding diffusion” the authors present an angle-based diffusion model for generating protein backbone structures.

The authors motivate the work in the following way:

"Other generative models rely on complex equivariant network architectures or loss functions to learn to generate a 3D point cloud that describes a protein structure [14, 15, 16, 17]. Such equivariant architectures can ensure that the probability density from which the protein structures are sampled is invariant under translation and rotation. However, translation- and rotation-equivariant architectures are often also symmetric under reflection, leading to violations of fundamental structural properties of proteins like chirality [15]."

However, the authors never argue why the equivariant network architectures being a complex is bad, or a problem, outside of the times when they are symmetric under reflection. However, they hedge by saying that they are “often” symmetric under reflection – what about those that are not? The authors go on to argue that “Intuitively, this point cloud formulation is detached from how proteins biologically fold – by twisting to adopt energetically favorable configurations [18, 19].” Why should it be a problem that the formulation is detached from folding? The authors’ method is also detached from folding, if considering the perspective of a biophysicist. Ultimately, I see no reason why a backbone generation model needs to involve or resemble a kinetic process – when it comes to generating backbone models, I’d much prefer a fast Monte Carlo method with near-zero rejection rate to a simulation of protein folding.

The authors demonstrate the method on proteins under 130 residues, often arbitrarily dividing it space into < 70 residues and > 70 residues. However, in the supplement they state that “even when considering longer structures up to 512 residues in length, the reconstructed structures still share a TMscore similarity much greater than 0.5 (graph not shown)”. It is hard to assess the usefulness of this model without seeing performance on “real” proteins, and “graph not shown” seems inappropriate when it could easily be added to the supplement.

As a reader, I immediately ask myself “can these backbone structures support side chains being built on top of them?” This issue is never addressed directly – in fact, the word “side-chain” is only present in the title of a reference and no form of the concept appears elsewhere. Are the difference between the backbone structures generated by the authors and the structures predicted from a protein design

process due the lack of consideration of side chains? Can this be investigated? There must be some sort of null model that can be used to understand this issue – for example, if the raw training structures are put through the same designability test, how would they fair? What about a set of OmegaFold structures?

Finally, could this model be extended to handle side chains? What about disulfides (another important component of protein structure that goes unmentioned)?

I think the paper needs moderate revision to the text and some additional controls (as described above) before it can be published in Nature Communications.

Reviewer #3 (Remarks to the Author):

The authors describe a novel approach for generating protein structures using backbone angles as input to a Transformer-Encoder trained via a diffusion process. After training, the network can denoise random noise to a sequence of valid backbone angles which can be used to reconstruct full 3D information. The manuscript is well written and authors make their code and model publicly available (thanks!). This work offers a comparatively simple solution to structure generation which avoids more complex equivariant networks and instead describes protein 3D structure using only relative orientations, i.e., backbone angles. However, there are some open questions on the experimental setup and to which extent the described evaluation allows to estimate generalisation of the proposed method.

You mention that FoldingDiff was trained for 10k steps but based on validation loss you picked the checkpoint after 1.4k steps due to poorer structure diversity. Did you apply the same logic to the auto-regressive (AR) model? - If not: maybe this explains the poorer structure diversity of the AR model?

This point leads me to my next concern: you mention in the manuscript that “[...] potential overlaps and similarities between training and test data does not artificially inflate any of the results we report.”. While I partly see your point here, I am still not fully convinced that this information leakage does not affect your work. While I agree that potential overlap between train and validation for determining early stopping won’t inflate your results, it might actually have the opposite effect and render your model worse than it could be by choosing a different checkpoint. After all, we know that protein sequences with 40% sequence similarity (your chosen similarity threshold) can still have highly similar structures. This also affects, in my eyes, your evaluation in Fig. 2b, 2c and 3a. While I understand that you do not use your test set proteins to directly compute metrics s.a. accuracy, you still use it to compare properties between generated and natural protein structures with natural proteins potentially having high structural overlap to your training set. This does not render your results completely invalid but, (again: in my eyes) unnecessarily lowers to which extent readers/users can expect your method to generalise to true out-of-distribution structures. I understand that changing this aspect of your work would require you to retrain your model from scratch but maybe there is a way to bypass this by only choosing proteins from your test set that have CATH homologous superfamilies (or better topologies) that do not appear in your training data. After all, you use CATH already and splitting based on CATH hierarchy is probably the best you can do for a good hold-out-test set in structure prediction (besides CASP). Such a change would add a lot of value to your evaluation as it would also render e.g. Fig. S3 more meaningful (otherwise, you

might end up measuring reconstruction error of structures that are highly similar to what you saw during training).

As the mentioned scTM score has certain shortcomings as discussed in the paper, assessing the quality of the generated structures in a different light could lead to interesting findings. One way could be to run one of the existing model accuracy estimators, s.a., DeepAccNet (<https://doi.org/10.1038/s41467-021-21511-x>) and put the predicted error of generated structures into perspective of predicted errors of natural test set structures.

For figures, consider adding side-chain information using e.g. <https://doi.org/10.1093/bioinformatics/btaa234> to make protein structures visually easier comparable (e.g. Fig. 4 where you compare your backbone against structures with side-chains).

Why did you distinguish between proteins with length/L <70? Instead (or additionally, both works) add another category which says <70, 70<128 and >128. This way readers could assess how well the method works for proteins longer than the training set cut-off (e.g. Fig. 4b.c).

For most examples you chose mostly alpha-helical structures. I think the manuscript would benefit from showing a protein that consists mostly of beta-sheets to make clear that this method also works in these cases.

Please add the number of parameters of all models (FoldingDiff as well as its autoregressive counterpart).

Did you experiment with other numbers of T=1k noising steps? - I thought this number is usually just a few hundred so I wondered whether you might have done any ablation on this or whether there was a particular motivation for this comparatively high value.

I would recommend being more careful with any comparison to the natural folding process. You already kept it vague by saying "inspired by" but maybe consider to make even more clear that the natural folding process is very different from FoldingDiff to avoid any misunderstanding.

In SOM you write "Even when considering longer structures up to 512 residues in length, the reconstructed structures still share a TMscore similarity much greater than 0.5 (graph not shown), which suggests that our method could scale up to larger structures." - Given that most users would like to apply your method to proteins with more than 128 residues, it would be valuable to understand the trade-off between length and accumulated error beyond your training cut-off. As you already have the data for this, please add it to the figure. The same holds true for Fig. S3: given that your model uses relative positional encoding, FoldingDiff should be able to generalise to protein lengths not seen during training. To give readers better understanding to which extent length affects usefulness of FoldingDiff, please also expand Fig S3 up to e.g. L=512. This also gives a nice additional data point to which extent relative positional encoding actually extrapolates for this problem/setup.

I think the manuscript would be more helpful for potential readers/users if the authors would provide a plot showing protein_length vs generation_time. This way, users could estimate how long generating e.g. 10 structures of a certain length takes.

The authors make the point that their method might capture protein dynamic and/or disorder. While I would understand if the authors were to leave this to future work, one way to easily quantify this would be to generate structures for the 117 proteins in this test set (<https://doi.org/10.1038/s41598-020-71716-1>) and correlate their residue-level structural fluctuation with the corresponding CheZOD-score. This would allow to put their performance directly into perspective of existing disorder predictors (and might give some support for the claim of capturing dynamics). But I consider the point on potential information leakage between train, test, val much more important, so I would understand if the authors leave this to future work (rather a suggestion to have a shortcut for assessment).

In the following point-by-point response, we have reproduced the reviewer's original comment in **bolded italics**, followed by normal text to indicate our response. Where appropriate, we reproduce passages from our main text inline to help highlight updates; in these cases, the updated text is shown in blue.

Response to reviewer #1

The authors propose a diffusion model named FoldingDiff for protein structure generation. FoldingDiff is a standard denoising diffusion probabilistic model that uses a generic transformer to model the backbone torsion and bond angles. They show that this can be used to generate protein backbones, about 22% of which are designable using inverse folding models such as ProteinMPNN. Progress in the field of ML-based protein structure modeling/design is extremely rapid now (also see below), and it is a highly exciting time (and the authors are certainly among those contributing to the progress and excitement); this also makes reviewing (as well as authoring) papers a bit more difficult. FoldingDiff was one of the earliest protein backbone diffusion models out in preprint, and the authors took their own, unique approach to the problem by focussing on a backbone description with angles, with its own advantages and disadvantages from which the field can learn. For this reason, I would be in favor of publication if the points below can be addressed.

We thank the reviewer for their positive assessment and feedback on our work and the suggestion for publication.

Major Comments

While FoldingDiff was out in preprint well ahead of models such as RFDiffusion, Chroma, Genie or FrameDiff, I do think it would be proper to cite and discuss at least RFDiff which is out now as a journal publication

We agree with the reviewer that more thorough discussion and citations for peer-reviewed, journal-published work on diffusion models for protein structures are appropriate. We have added reference to RFDiffusion and Genie to the manuscript Introduction as follows:

Other generative models rely on complex equivariant network architectures or loss functions to learn to generate a 3D point cloud that describes a protein structure [14, 15, 16, 17, 18, 19]. Of notable success among these methods is RFDiffusion [18], which not only presents a myriad of conditional generation applications that can design proteins binding specific targets, but also performs thorough experimental validation of computationally generated proteins.

Since torsion/bond angles are invariant representations that only implicitly capture its 3D localization, I am curious how well the model learns to prevent atomic clashes compared to point

cloud or AF2 frame-based models. Please provide some quantification of atomic clashes (ex. % of atoms < literature atomic radii) and comparisons to other models.

We thank the reviewer for pointing out this important analysis point, and we agree that providing this metric would add value to our work. As such, we have analyzed the frequency of backbone clashes for 3 sets of structures: naturally-occurring CATH domains, FoldingDiff's generated backbones, and RFDiffusion's generated backbones. The text and table describing these results has been added as a new subsection and new Table S1 within the "Angular representations of proteins" section of our supplementary material, and are reproduced below for ease of reference:

Another concern when adopting an angular view of protein structures is atomic clashes, which can arise when a series of angles generated by FoldingDiff results in molecules that clash in Cartesian space. To understand how often these occur, we analyze clash frequencies for three sets of proteins: natural CATH structures between 50 and 128 residues in length, FoldingDiff generations, and RFDiffusion [18] generations. Both RFDiffusion and FoldingDiff were asked to unconditionally generate 10 structures for each length in the range $l \in [50, 128)$ for a total of 780 structures each. For this analysis, we consider two non-consecutive backbone atoms with Van der Waal radii x, y to be clashing if their pairwise Euclidean distance is less than $0.63(x + y)$, indicating a large overlapping volume. Distances for Van der Waal radii are taken from Bondi [70].

The frequency of clashes and median clashes per structure are shown in Table S1. We observe that FoldingDiff exhibits significantly more clashes than both natural CATH structures and structures generated by RFDiffusion. However, these can be easily filtered out, or passed through structural refinement methods to resolve clashes. Within our main results, we present all structures without such refinement or filtering, such as to most directly depict FoldingDiff's capabilities.

And the referenced table and corresponding caption:

Category	Total structures	Clash-free structures	Median clashes per structure
CATH	13963	13720 (98.3%)	0
FoldingDiff (ours)	780	364 (46.7%)	2
RFDiffusion [67]	780	768 (98.5%)	0

Number of atomic clashes observed within CATH structures, FoldingDiff generations, and RFDiffusion generations of comparable length.

On the issue of error accumulation (line 91-96): I don't think this is the proper way to test whether the lever effect is not an issue, since all this is doing is going from native 3D coordinates to internal coordinates and back. If some error is introduced by a model and the internal coordinates are imperfect, it will likely affect the downstream coordinates significantly in terms of RMSD/TM. Thus,

It would be interesting to see the reconstruction error when the angles are perturbed by increasing levels of Gaussian noise, and observe how robust this representation is to noise.

We think this suggested analysis would lend great value to our work. We have performed additional experiments to more directly demonstrate the impact of the lever effect, and added new text and the new Figure S4 as a new section in the “Angular representation of proteins” supplementary section, which we point readers to when discussing limitations in our discussion section. Please see the relevant text reproduced below:

A key concern when adopting an angular description of protein structures is the potential for single-angle errors to drastically alter the global structure – a lever arm effect. To test this empirically, we take a CATH structure’s angular representation a , and then perturb one instance of its constituent dihedral angles (i.e., ψ , ϕ , ω in our main text) by a small amount, thus creating \hat{a} . We then calculate the TMscore between the structures specified by a and \hat{a} (Figure S4). We perform a set of perturbations proportional to natural angular variance, and a set of perturbations proportional to angular differences introduced by FoldingDiff’s reconstruction of partially noised test set structures (250 steps, Figure S3).

We repeat this analysis for three randomly selected CATH structures, perturbing each of the three dihedrals for each position in the set of $N - 1$ angles specifying the backbone (Figure S4). Despite a and \hat{a} having only one “edit” between them, we observe that even single errors proportional to the native data distribution (Figure S4, left column) can have an appreciable impact on overall structure, especially when they occur in the center of the protein. In comparison, perturbations proportional to FoldingDiff’s observed distribution of reconstruction differences result in much less drastic changes in overall structure (Figure S4, right column), suggesting that FoldingDiff is relatively robust to these lever arm effects at this error scale. Nonetheless, we still observe the same trend as with larger perturbations, namely that, controlling for magnitude, errors occurring in central positions have the greatest effects on overall structure.

Please also see the new, referenced figure below along with its caption:

Figure S4: **Empirical demonstration of lever arm effects on three CATH structures.**

For each of three randomly selected CATH structures (a-c), TMscore comparison of the “correct” structure specified by the original series of six angles describing protein backbones (see Table 1), to a “perturbed” structure specified by a set of angles including a single perturbed dihedral (ϕ , ψ , ω ; blue, orange, green lines respectively). The relevant dihedral was perturbed by either 0.5 times its natural standard deviation ($0.5\sigma_{data}$; left column), or 0.5 times the standard deviation in angular difference arising when FoldingDiff reconstructs that angle from partially noised test structures ($0.5\sigma_{test}$), offset by the average difference ($\mu_{test} +$

$0.5\sigma_{test}$; right column). This perturbation was applied to each of the $N - 1$ positions along the structure, for each of the 3 dihedral backbone torsional angles. Each plot shows the change in TMscore (y-axis) as a function of the perturbed dihedral (colored lines) and the position of the perturbation (x-axis).

We have acknowledged the presence of this lever arm effect and discussed it in the Supplementary sections as indicated above. As supported by the range of analyses and evaluations we present in our manuscript, we note that FoldingDiff's generated structures are robust – as measured by our angle distribution, secondary structure, and TMscore metrics – despite the presence of this lever arm effect. We believe this new analysis addresses the reviewer's concerns surrounding our representation's robustness, and demonstrates one of the primary drawbacks of our approach.

Minor Comments

Since FoldingDiff predicts the omega torsion angle which in turn learn the correct placement of oxygen atoms, I am not sure why pyMOL fails to annotate secondary structures in the protein structures, which should default to proper secondary structure visualization given correct hydrogen bonding.

As the reviewer notes, according to PyMOL's documentation, PyMOL's secondary structure annotation method "dss" uses a combination of hydrogen bonding patterns and geometry to annotate secondary structures. As side chains are significantly involved in aforementioned hydrogen bonding interactions, a structure without side chains (which is FoldingDiff's output) would be missing this critical information, implying that PyMOL would not annotate our generated structures' secondary structures even given correct backbone geometry.

We can empirically demonstrate this by taking a .pdb structure that PyMOL annotates with appropriate secondary structure, and re-visualizing it after merely removing side chain atoms. Below, we show such an original structure on the left, and the structure without side-chains on the right. PyMOL no longer shows alpha helix ribbon cartoons on the right.

Nonetheless, we agree that secondary structure visualization would greatly aid comprehension and impact of our work. To facilitate this, we have used the third-party tool PSEA (which we already use elsewhere in our work for secondary structure analysis) to annotate secondary structures within FoldingDiff's generations, which we then pass to PyMOL to illustrate. Notably, this change only applies to visualizations of "raw" generations without amino acid information; visualizations of structures produced by OmegaFold, for example, are unchanged. As examples, see the below reproductions of Figures 4a and 4d; Figure panels 1a, 1d, and 2a have been revised with minor updates in this spirit as well.

Figure 4a:

Figure 4d:

We have also added an additional Methods section entitled "3D visualization of protein structures" to clearly delineate how we annotate and illustrate proteins; please refer to the relevant text reproduced below:

3D visualization of protein structures

3D visualizations of protein structures are done via PyMOL. For CATH structures and structures generated by AlphaFold2 and OmegaFold, secondary structure cartoons are drawn based on annotations from PyMOL's built-in "dss" method. Structures generated by FoldingDiff do not have side chain information, and thus do not contain the hydrogen bonding information required for "dss" to work properly. To illustrate FoldingDiff's final generations, we instead draw secondary structures as annotated by P-SEA, which we also use throughout our manuscript for secondary structure evaluation. We additionally note that oxygen atoms must be present for PyMOL's cartoons to display properly; thus, we insert oxygen atoms in our generated backbone structures (which canonically include only N-Ca-C atoms) in a coplanar, trans configuration with respect to the peptide bond.

Backbone diversity visualization (line 175-181): A UMAP projection solely of generated samples seems less informative – it would be interesting to add native structures to the projections and analyze the differences in coverage of the various descriptors.

This is a great suggestion to improve our interpretation of FoldingDiff's sampled protein backbones. We have included a new set of UMAP visualizations that jointly embeds FoldingDiff's generated backbones as well as all test set structures between [50, 128) residues in length. These UMAP plots, reproduced below, have been added as the new Figure S10 to help readers contextualize how our generations compare with natural structures, along with the accompanying caption.

Visualization of Gauss integral embeddings and structural features of FoldingDiff generations and natural CATH structures. (a) UMAP visualization of Gauss integral embeddings for natural test-set ($n=847$, blue) structures between 50 and 128 residues in length, and structures generated by FoldingDiff ($n=780$, orange). These same embeddings are additionally colored by the number of alpha helices (b) and beta sheets (c) as detected by P-SEA. In panels (b-c), marker types denote whether each point corresponds to a CATH (circle) or FoldingDiff (cross) example.

We have additionally added the following text to our results section "Structural characterization of FoldingDiff generations" to incorporate our observations from this plot:

.... We annotated this plot according to several descriptors -- length, number of helices, and number of sheets -- and observed that the design space of generated backbones spans a large range of these descriptors (Figures 3d-f). Jointly visualizing these embeddings in the context of CATH test set structures of similar length (Figure S10) reveals that FoldingDiff's generations share regions of overlap with natural structures while also exploring embedding spaces only sparsely populated by natural structures. This suggests that FoldingDiff has the potential to sample backbones occupying a range of similarities to known structures.

Baseline (line 190-195): I am honestly not sure how valid this baseline is – it seems pretty obvious that this baseline will not generate proper protein structures even if the angles may be sampled from the proper angular distribution. The autoregressive baseline seems sufficient.

We agree with the reviewer in their assessment that this “random sampling” analysis would not be expected to generate proper structures. Our motivation for including this analysis is twofold - firstly to show concretely that correct angular distributions alone are insufficient to yield reasonable structures, and secondly to establish a sort of “lower bound” for performance. Therefore, we retain this analysis along with the autoregressive baseline.

Designability (line 254-256): One concern I have is that a lower distribution of training TM-scores may be due to FoldingDiff not generating unrealizable protein structures, which can be argued given the ~22% scTM scores. Do structures with scTM < 0.5 at least visually resemble realistic structures? It would also be nice to see some randomly selected samples at various scTMs in the supplementary.

To help readers better visualize and understand the range of scTM values and what sorts of structures generally fall into each range, we have added a new supplementary Figure S17 that illustrates 4 randomly chosen structures for each scTM range between: [0, 0.25), [0.25, 0.5), [0.5, 0.75), and [0.75, 1.0). The added manuscript text discussing this figure is as follows:

.... In addition, OmegaFold's predicted folds (middle row) are consistently very similar to FoldingDiff's initial generated backbone in each of these cases, qualitatively suggesting that designability also may be greater than scTM scores themselves might suggest. Figure S17 additionally shows randomly selected structures spanning the entire range of scTM designability scores. Overall, we observe that structures with very poor designability (scTM < 0.25) tend to include long, unstructured loop regions connecting interspersed regions of beta sheets. Structures with high designability (scTM \geq 0.75) are not very diverse and tend to incorporate mostly alpha helices with minimal, if any, kinks and turns. Structures in the middle of this range ($0.25 \leq$ scTM < 0.75) appear qualitatively reasonable and contain a strong variety of combinations of secondary structure motifs.

The figure and accompanying caption are reproduced below:

Figure S17: Representative FoldingDiff generations spanning the range of scTM designability scores. From top to bottom, each row shows structures of increasing designability with the ranges [0.0, 0.25), [0.25, 0.5), [0.5, 0.75), [0.75, 1.0). Within each row, structures are sorted in increasing designability from left to right. Structures are colored in a rainbow hue from N to C terminus.

Response to reviewer #2

In "Protein structure generation via folding diffusion" the authors present an angle-based diffusion model for generating protein backbone structures. The authors motivate the work in the following way: "Other generative models rely on complex equivariant network architectures or loss functions to learn to generate a 3D point cloud that describes a protein structure [14, 15, 16, 17]. Such equivariant architectures can ensure that the probability density from which the protein structures are sampled is invariant under translation and rotation. However, translation- and rotation-equivariant architectures are often also symmetric under reflection, leading to violations of fundamental structural properties of proteins like chirality [15]."

We thank the reviewer for their constructive criticism and feedback on our work, which we have used to improve and refine our results and exposition.

.... However, they hedge by saying that they are "often" symmetric under reflection - what about those that are not? The authors go on to argue that "Intuitively, this point cloud formulation is detached from how proteins biologically fold - by twisting to adopt energetically favorable configurations [18, 19]." Why should it be a problem that the formulation is detached from folding? The authors' method is also detached from folding, if considering the perspective of a biophysicist. Ultimately, I see no reason why a backbone generation model needs to involve or resemble a kinetic process - when it comes to generating backbone models, I'd much prefer a fast Monte Carlo method with near-zero rejection rate to a simulation of protein folding.

We appreciate the reviewer providing thoughtful critique of our problem formulation. The reviewer is correct in pointing out that our method is also "detached" from folding, in that we do not perform energetics calculations, for example, to perform diffusion update steps. Our intent is to draw *inspiration* from the biological folding process - not to simulate it - as a means to construct a natural problem formulation. The text in our initial submission clearly states this, and we have revised the text to further emphasize that we draw inspiration from the folding process (and are not directly simulating it). Please see the following reproduced text segments.

In the third paragraph of our introduction:

.... Combining these ideas, our framework generates novel backbones by starting from a set of random angles that correspond to a random, unfolded state and iteratively denoising the underlying angles to arrive at a final backbone structure (Figure 1). Although this angular denoising procedure does not directly capture any biophysical folding processes, it draws inspiration from how proteins twist and fold into their final structures; as such, we name our method FoldingDiff.

In the first paragraph of our results section "Designing and training FoldingDiff":

.... After training is complete, to generate new data points, the diffusion model starts from random noise and applies T steps of iterative denoising where the output of each prior denoising step is used to prepare the input for the next cycle of denoising, culminating in a “clean” sample (Algorithm 1, Figure 1d). **Importantly, this noising and denoising procedure does not approximate any biophysical processes of protein folding.**

We believe that having a problem formulation and protein generative model that is biologically motivated has key advantages:

1. Constrained solution space: In a naive formulation whereby each backbone atom is assigned a Cartesian coordinate, nearly all possible configurations of the resulting point cloud are structurally impossible, as the points are far too distant to form requisite bonds. On the other hand, with our angle based formulation, structurally impossible configurations are effectively ruled out by construction (barring clashes), as many configurations produced by our model are likely physically valid even if they are energetically unfavorable. As a result, our biologically motivated formulation effectively reduces the solution space relative to a point cloud formulation, which may lead to more efficient learning. Indeed, we see empirical evidence of this in our experiment replacing our angular formulation with Cartesian coordinates and training the same transformer network – despite this baseline and FoldingDiff having identical architectures and identical amounts of training data, FoldingDiff performs much better, likely in part due to its solution space being more constrained.
2. Principled sampling from intermediate diffusion timesteps: A common usage of diffusion models is to perform “partial denoising” starting from an intermediate timestep to sample data points that are related to each other. Under a Cartesian formulation, this intermediate timestep typically corresponds to a completely disjointed point cloud. Under our angular formulation, this intermediate timestep typically corresponds to a “semi-folded” structure that has twists and angles suggestive of secondary structure, but that have yet to coalesce into definite configurations. We hypothesize that performing partial denoising from a structure from an intermediate timestep may yield a better ensemble of related structures than denoising from a partially configured point cloud; this motivates future investigation that could lend utility in molecular simulations and other applications.

With respect to the relationship of FoldingDiff to more complex models or architectures (e.g., equivariant networks), we point to the large body of work exploring physics-constrained machine learning models, as well as the community of researchers working on developing models with inductive priors to fit a problem space. The literature in these fields has repeatedly demonstrated the utility of models incorporating inductive priors in their design. These efforts often yield greater efficiency and expressivity compared to larger, more complex models that may exhibit gains in raw performance, but may suffer from inefficiencies and difficulties with generalization.

FoldingDiff provides a problem formulation and modeling framework that lends greater efficiency (i.e., by constraining the solution space and leveraging a vanilla transformer background) and

expressivity (i.e., via principled sampling from intermediate diffusion steps), which we believe are valuable to the protein design, computational biology, and greater machine learning communities.

The authors demonstrate the method on proteins under 130 residues, often arbitrarily dividing it space into < 70 residues and > 70 residues. However, in the supplement they state that “even when considering longer structures up to 512 residues in length, the reconstructed structures still share a TMscore similarity much greater than 0.5 (graph not shown)”. It is hard to assess the usefulness of this model without seeing performance on “real” proteins, and “graph not shown” seems inappropriate when it could easily be added to the supplement.

When evaluating our generated proteins for designability via scTM scores, we originally divided our structures into those < 70 residues and those > 70 residues following what prior works have done (Trippe et al.). We agree that the threshold of 70 residues is somewhat arbitrary. In light of this, we have conducted a new analysis evaluating the distribution of scTM scores as a function of length, added to the manuscript as the new Figure S14 and reproduced below, along with its caption. We observe a significant negative correlation between length and designability.

Distribution of scTM scores by generated structure length. Self-consistency TM score (scTM, y-axis), evaluated with ProteinMPNN and OmegaFold, versus length (x-axis) for all of FoldingDiff's generated proteins. The solid line indicates the average scTM at each length, with the shaded region indicating range between the highest and lowest observed scTM for that length. Dotted gray horizontal line indicates scTM

cutoff for designability at 0.5. There is a significant correlation between the two, with longer proteins exhibiting lower designability scTM scores (Spearman's $\rho = -0.38$, $p = 3.12 \times 10^{-28}$).

This discussion is also newly incorporated into the manuscript results section as follows:

Compared to this prior work, FoldingDiff improves the designability of both short sequences (up to 70 residues, 76/210 designable compared to 36/210, $p = 1 \times 10^{-5}$, Chi-square test) and long sequences (beyond 70 residues, 87/570 designable compared to 56/570, $p = 5.6 \times 10^{-3}$, Chi-square test). Despite these consistent improvements in designability relative to ProtDiff, we similarly find that longer generated structures tend to have poorer designability (Spearman's $\rho = -0.38$, $p = 3.12 \times 10^{-28}$, Figure S14). While ProteinSGM...

We clarify that the referenced discussion of long structures up to 512 residues in length was exclusively in the context of converting backbones into angular representation and back - i.e., not for generative modeling. The FoldingDiff model itself is only trained and evaluated up to 128 residues, as noted clearly in our main text, lines 132-135 and 140-142. To further clarify this distinction, we have updated our supplemental text as follows:

This indicates that while our representation itself is slightly "lossy", the losses do not change the overall fold. Even when converting longer structures up to 512 residues in length to an angular representation and back, the reconstructed structures still share a TMscore similarity much greater than 0.5 (graph not shown), which suggests that our method could scale up to larger structures. Note, however, that our generative FoldingDiff model is currently only trained on structures up to 128 residues in length -- this analysis focuses solely on the potential for errors introduced by our angular representation, and not FoldingDiff's ability to work effectively within this formulation or to design structures longer than 128 residues in length.

As a reader, I immediately ask myself "can these backbone structures support side chains being built on top of them?" This issue is never addressed directly - in fact, the word "side-chain" is only present in the title of a reference and no form of the concept appears elsewhere. Are the difference between the backbone structures generated by the authors and the structures predicted from a protein design process due the lack of consideration of side chains? Can this be investigated? There must be some sort of null model that can be used to understand this issue - for example, if the raw training structures are put through the same designability test, how would they fair? What about a set of OmegaFold structures? Finally, could this model be extended to handle side chains? What about disulfides (another important component of protein structure that goes unmentioned)?

FoldingDiff is a method for generating protein *backbone* structures – an outstanding problem in structure-based protein design, where the paradigm is to first design a backbone structure, and then infer or design an amino acid sequence (e.g., specifying side chains) that will fold into that structure. Thus, by definition FoldingDiff does not consider side chain information. This point is clearly stated in many places as we consistently describe FoldingDiff as a method for generating backbones. To further clarify, we have made the following update in the manuscript text:

These six angles are defined precisely in Table 1 and illustrated in Figure 1a. Notably, these angles do not specify side chain identity or orientation; like other works tackling backbone structure generation, FoldingDiff focuses on designing backbones and relies on external methods to subsequently infer amino acids that fold into designed structures. These internal angles can be easily computed....

Furthermore, in the scTM designability evaluation, we take generated structures from FoldingDiff and infer amino acid sequences that would fold into the generated backbone structure via inverse folding methods like ProteinMPNN (Dauparas et al., 2022). We then use OmegaFold/AlphaFold2 to predict structures for these amino acid sequences and compare their consistency to FoldingDiff's original generations. If one were interested instead in interrogating what FoldingDiff's generations would look like with ProteinMPNN's predicted amino acid side chains (rather than directly folding the sequence), one could graft side chains onto FoldingDiff's generated backbone using methods like FASPR.

Regarding a null model, our original submission includes a null model for designability, described in the "Designability of generated backbones" results section, line 257 as follows.

.... Conversely, we evaluate experimental structures to establish an upper bound for designability. 87% of natural structures have an scTM ≥ 0.5 . (Figure S16). The fact that even real structures do not have perfect designability demonstrates that the scTM evaluation pipeline can be fragile, and that some fraction of FoldingDiff's generations may produce interesting, foldable proteins even if the scTM pipeline does not identify them as such.

For this null model, we run the scTM designability pipeline on natural CATH domains and use the values to contextualize the scTM scores for our generated backbones. It does not appear to be informative to run OmegaFold structures through an scTM pipeline, as OmegaFold takes in an amino acid sequence and predicts a structure (like AlphaFold2), which is a fundamentally different problem than the generative task we are trying to address in our work.

Regarding extensions to handle side chains, we previously highlighted this direction in our discussion. We have revised the corresponding text to more explicitly discuss side chain generation; please refer to the reproduced text below:

Our generated structures are still of relatively short lengths compared to natural proteins, which typically have several hundred residues; future work could extend towards longer structures, potentially incorporating additional losses or inputs that help “checkpoint” the structure and reduce accumulation of error. Future work could build upon FoldingDiff’s backbone generation functionality to additionally perform amino acid sequence and side chain conformation generation, rather than relying on external inverse folding methods.

Modeling disulfide bonds may be possible, but doing so seems to be more in the domain of methods that predict protein folds from structures, and less within the domain of generative models of protein structure. It is possible that a future iteration of FoldingDiff that additionally models side chain information (as discussed above) might capture disulfide bond information, but we would like to reiterate that this is pending future research efforts.

We hope these clarifications address the reviewer’s questions regarding side chains.

Response to reviewer #3

The authors describe a novel approach for generating protein structures using backbone angles as input to a Transformer-Encoder trained via a diffusion process. After training, the network can denoise random noise to a sequence of valid backbone angles which can be used to reconstruct full 3D information. The manuscript is well written and authors make their code and model publicly available (thanks!). This work offers a comparatively simple solution to structure generation which avoids more complex equivariant networks and instead describes protein 3D structure using only relative orientations, i.e., backbone angles. However, there are some open questions on the experimental setup and to which extent the described evaluation allows to estimate generalisation of the proposed method.

We thank the reviewer for their positive assessment of our work’s motivation and execution, as well as for their thoughtful critique.

You mention that FoldingDiff was trained for 10k steps but based on validation loss you picked the checkpoint after 1.4k steps due to poorer structure diversity. Did you apply the same logic to the auto-regressive (AR) model? - If not: maybe this explains the poorer structure diversity of the AR model?

Thank you for raising this detailed question. For the AR baseline, we observed that the loss on the validation set did not improve after ~1.4k epochs; accordingly, we used the corresponding checkpoint for all autoregressive model generations. We apologize that this detail was omitted in the initial submission, and we have updated our Methods text accordingly.

.... We train f_{AR} using the AdamW optimizer with weight decay set to 0.01. We use a

batch size of 256 over 10,000 epochs, linearly scaling the learning rate from 0 to 5×10^{-5} over the first 1,000 epochs, and back to 0 over the remaining 9,000 epochs. We find that the validation loss does not improve beyond 1,446 epochs of training, and use this model checkpoint for generation.

This point leads me to my next concern: you mention in the manuscript that “[...] potential overlaps and similarities between training and test data does not artificially inflate any of the results we report.”. While I partly see your point here, I am still not fully convinced that this information leakage does not affect your work. While I agree that potential overlap between train and validation for determining early stopping won’t inflate your results, it might actually have the opposite effect and render your model worse than it could be by choosing a different checkpoint. After all, we know that protein sequences with 40% sequence similarity (your chosen similarity threshold) can still have highly similar structures. This also affects, in my eyes, your evaluation in Fig. 2b, 2c and 3a. While I understand that you do not use your test set proteins to directly compute metrics s.a. accuracy, you still use it to compare properties between generated and natural protein structures with natural proteins potentially having high structural overlap to your training set. This does not render your results completely invalid but, (again: in my eyes) unnecessarily lowers to which extent readers/users can expect your method to generalise to true out-of-distribution structures. I understand that changing this aspect of your work would require you to retrain your model from scratch but maybe there is a way to bypass this by only choosing proteins from your test set that have CATH homologous superfamilies (or better topologies) that do not appear in your training data. After all, you use CATH already and splitting based on CATH hierarchy is probably the best you can do for a good hold-out-test set in structure prediction (besides CASP). Such a change would add a lot of value to your evaluation as it would also render e.g. Fig. S3 more meaningful (otherwise, you might end up measuring reconstruction error of structures that are highly similar to what you saw during training).

We appreciate the reviewer’s proposal of a unique evaluation of our generative model. It is certainly true that protein sequences with low sequence similarity can still share similar structures. Nonetheless, we do not believe that using our randomized test set (with aforementioned 40% protein sequence similarity cutoff) compromises FoldingDiff’s capabilities, nor does it decrease the validity of our evaluations.

Perhaps most saliently, generative models (and more broadly, deep learning models in general) are known to struggle when generalizing to true out of distribution examples. After all, one would not expect a generative model trained on structures of mostly alpha helices to be able to generate structures of mostly beta sheets – it simply has not seen enough examples of beta sheet rich structures. With this in mind, we think an appropriate goal is to train a generative model, such as FoldingDiff, on a diverse range of examples, and evaluate if it is able to subsequently sample a similarly diverse range of examples. Indeed, this goal of diverse, high-quality sampling motivates much of the evaluations we present, from secondary structure to designability analyses. These analyses suggest the existence of at least some structures with high scTM (designability) and low

similarity to the diverse set of training structures, which we believe is a strong evaluation of FoldingDiff's ability – as a generative model – to generalize and explore novel structural folds. More broadly, while careful test set evaluations are critical to evaluating supervised predictive models – as their primary purpose is to predict robustly on new, unseen examples – we believe that a generative model's aim to sample new data is best assessed through direct evaluation of its generations.

As the mentioned scTM score has certain shortcomings as discussed in the paper, assessing the quality of the generated structures in a different light could lead to interesting findings. One way could be to run one of the existing model accuracy estimators, s.a., DeepAccNet (<https://doi.org/10.1038/s41467-021-21511-x>) and put the predicted error of generated structures into perspective of predicted errors of natural test set structures.

We agree that our work can benefit from additional analyses surrounding the accuracy of folding models and thank the reviewer for suggesting this direction. Unfortunately, we were unable to configure the DeepAccNet method to run successfully on our machines, despite following the authors' posted instructions on their GitHub page and trying various different configurations (both the PyTorch and TensorFlow implementations). In lieu of using an external method to estimate model accuracy, we instead perform additional analysis characterizing the relationship between scTM and pLDDT confidence scores from the protein folding methods OmegaFold and AlphaFold2. pLDDT reflects OmegaFold's/AlphaFold2's confidence in its structure prediction for each residue. Please see the following figure (and caption), which has been newly added as Figure S15:

scTM designability scores for FoldingDiff's generations as a function of protein folder confidence. For each structure generated by FoldingDiff, ProteinMPNN [47] was used to sample 8 candidate amino acid sequences likely to realize the generated structure and OmegaFold [48] (a) or AlphaFold2 [51] (b) was then used to computationally fold and validate these sequences. The scTM designability score (y-axis) was computed using the best match among the 8 candidates. Both OmegaFold

(a) and AlphaFold2 (b) also produce predicted local-distance difference test (pLDDT) scores per residue, indicating confidence in their predicted folds. The average pLDDT (x-axis) for the closest-matching predicted structure for each of FoldingDiff's generations (i.e., the structure yielding scTM scores) was compared to scTM (y-axis). There is a significant positive correlation between pLDDT and scTM for both OmegaFold (Spearman's $\rho = 0.56$, $p = 4.61 \times 10^{-65}$) and AlphaFold2 (Spearman's $\rho = 0.47$, $p = 1.42 \times 10^{-44}$). Dotted horizontal line indicates 0.5 scTM threshold for designability.

We refer to this new result and figure in our results section "Designability of generated backbones". Please refer to the following reproduced text:

.... Therefore, the designability reported in ProteinSGM does not directly reflect its generative process, as Rosetta post-processing significantly improves the viability of their structures. We also find that confidence scores produced by OmegaFold and AlphaFold2 correlate well with scTM (Figure S15), suggesting that applications leveraging FoldingDiff could directly use these scores to identify high-quality generations.

For figures, consider adding side-chain information using e.g. <https://doi.org/10.1093/bioinformatics/btaa234> to make protein structures visually easier comparable (e.g. Fig. 4 where you compare your backbone against structures with side-chains).

We thank the reviewer for pointing us towards resources to help clarify our visualizations. In regard to the protein structures illustrated in Figure 4, these originally lacked the cartoon ribbons highlighting helices and sheets due to PyMOL not annotating them, and indeed this makes the structures more difficult to visually compare. To rectify this, we are now taking the PSEA annotations originally done to quantify secondary structure content and using them to ask PyMOL to explicitly draw in these cartoons. Updated panels are reproduced below for ease of reference.

Figure 4a:

Figure 4d:

Figures 1a, 1d, and 2a have seen minor updates in the same fashion. We have also added a new Methods section titled “3D visualization of protein structures” to clarify how these illustrations are generated; please see the following reproduced text:

3D visualization of protein structures

3D visualizations of protein structures are done via PyMOL. For CATH structures and structures generated by AlphaFold2 and OmegaFold, secondary structure cartoons are drawn based on annotations from PyMOL's built-in “dss” method. Structures generated by FoldingDiff do not have side chain information, and thus do not contain the hydrogen bonding information required for “dss” to work properly. To illustrate FoldingDiff's final generations, we instead draw secondary structures as annotated by P-SEA, which we also use throughout our manuscript for secondary structure evaluation. We additionally note that oxygen atoms must be present for PyMOL's cartoons to display properly; thus, we insert oxygen atoms in our generated backbone structures (which canonically include only N-Ca-C atoms) in a coplanar, trans configuration with respect to the peptide bond.

Why did you distinguish between proteins with length/L <70? Instead (or additionally, both works) add another category which says <70, 70<128 and >128. This way readers could assess how well the method works for proteins longer than the training set cut-off (e.g. Fig. 4b.c).

We appreciate the reviewer's careful critique. We choose to distinguish between proteins above and below 70 residues in length to be directly comparable to work previously done by the ProtDiff team

(Trippe et al.). Nonetheless, we agree that the 70 residue threshold may be a bit arbitrary, and have added a new supplementary Figure S14 to more directly interrogate the relationship between generation length and scTM. Please see the updated figure below:

Distribution of scTM scores by generated structure length. Self-consistency TM score (scTM, y-axis), evaluated with ProteinMPNN [47] OmegaFold [48], versus length (x-axis) for all of FoldingDiff's generated proteins. The solid line indicates the average scTM at each length, with the shaded region indicating range between the highest and lowest observed scTM for that length. Dotted gray horizontal line indicates scTM cutoff for designability at 0.5. There is a significant correlation between the two, with longer proteins exhibiting lower designability scTM scores (Spearman's $\rho = -0.38$, $p = 3.12 \times 10^{-28}$).

As well as the corresponding text added to our "Designability of generated backbones" results:

Compared to this prior work, FoldingDiff improves the designability of both short sequences (up to 70 residues, 76/210 designable compared to 36/210, $p = 1 \times 10^{-5}$, Chi-square test) and long sequences (beyond 70 residues, 87/570 designable compared to 56/570, $p = 5.6 \times 10^{-3}$, Chi-square test). Despite these consistent improvements in designability relative to ProtDiff, we similarly find that longer generated structures tend to have poorer designability (Spearman's $\rho = -0.38$, $p = 3.12 \times 10^{-28}$, Figure S14). While ProteinSGM...

To be clear, we do not make any claims around our model's ability to generate structures beyond 128 residues in length. This misunderstanding may have arisen because in our supplementary information, we discuss the reconstruction of proteins up to 512 residues in length. However, this is solely focused on the effects of translating structures from coordinate-based to angle-based representations and back, and does not pertain to generative modeling. We apologize for any confusion, and have updated our supplemental text to be clearer about this point. See the below for the added text:

... the reconstructed structures still share a TMscore similarity much greater than 0.5 (graph not shown), which suggests that our method could scale up to larger structures. *Note, however, that our generative FoldingDiff model is currently only trained on structures up to 128 residues in length – this analysis focuses solely on the potential for errors introduced by our angular representation, and not FoldingDiff's ability to work effectively within this formulation or to design structures longer than 128 residues in length.*

For most examples you chose mostly alpha-helical structures. I think the manuscript would benefit from showing a protein that consists mostly of beta-sheets to make clear that this method also works in these cases.

With the aforementioned update to Figure 4 to correctly show PyMOL cartoon illustrations of secondary structure, we believe that Figure 4 now better showcases cases of beta sheets being present in FoldingDiff's generations, which were difficult to visually identify before. That being said, it is difficult to find examples of generated structures that consist of mostly beta sheets. Indeed, further analysis reveals that of the 177 designable backbones from FoldingDiff, only 16 contain beta sheets, whereas out of the remaining 603 backbones with scTM < 0.5, 533 contain beta sheets. This indicates that FoldingDiff has a relatively difficult time generating designable structures (as measured by scTM) that contain beta sheets - keeping in mind that even natural structures are not necessarily designable, as defined by scTM > 0.5. We have updated our "Designability of generated backbones" results section with the following text to highlight this result:

...making these structures biologically uninteresting even if they might be designable. *Finally, we find that of our 177 backbones considered to be designable with OmegaFold, only 16 contain beta sheets as annotated by P-SEA. Conversely, of the remaining 603 backbones with scTM < 0.5, 533 contain beta sheets. This suggests that FoldingDiff may have greater difficulty generating high-scTM structures with beta sheets ($p \ll 10^{-5}$, Chi-square test). Although we believe that some degree of this is due to the fragility of the scTM pipeline itself, there is certainly much work that can be done to improve FoldingDiff's ability to model complex arrangements of beta sheets.*

We agree that additional examples would be useful for the reader. As such, we have added a new supplementary figure that illustrates randomly chosen examples across the range of scTM values. This figure shows several examples of structures of mainly beta sheets, and is reproduced below for ease of reference:

.... In addition, OmegaFold's predicted folds (middle row) are consistently very similar to FoldingDiff's initial generated backbone in each of these cases, qualitatively suggesting that designability also may be greater than scTM scores themselves might suggest. Figure S17 additionally shows randomly selected structures spanning the entire range of scTM designability scores. Overall, we observe that structures with very poor designability (scTM < 0.25) tend to include long, unstructured loop regions connecting interspersed regions of beta sheets. Structures with high designability (scTM \geq 0.75) are not very diverse and tend to incorporate mostly alpha helices with minimal, if any, kinks and turns. Structures in the middle of this range (scTM between 0.25 and 0.75) appear qualitatively reasonable and contain a strong variety of combinations of secondary structure motifs.

The figure and accompanying caption are reproduced below:

Figure S17: Representative FoldingDiff generations spanning the range of scTM designability scores. From top to bottom, each row shows structures of increasing designability with the ranges [0.0, 0.25), [0.25, 0.5), [0.5, 0.75), [0.75, 1.0). Within each row, structures are sorted in increasing designability from left to right. Structures are colored in a rainbow hue from N to C terminus.

Please add the number of parameters of all models (FoldingDiff as well as its autoregressive counterpart).

FoldingDiff has 14.56M parameters; the autoregressive model has slightly fewer (14.41M) parameters due to its usage of a different positional embedding that yielded improved validation loss. These have been included in the Methods as follows:

In our “Modeling and datasets” section describing our primary FoldingDiff model:

.... To convert the transformer’s final per-position representations to our six outputs, we apply a regression head consisting of a densely connected layer, followed by GELU activation [62], layer normalization, and finally a fully connected layer outputting our six values. In total, our model has 14.56 million trainable parameters.

Similarly, in our “Autoregressive baseline model” section:

The total length of the sequence is encoded using random Fourier feature embeddings, similarly to how time was encoded in FoldingDiff, and this embedding is similarly added to each position in the sequence of angles. The model, which consists of 14.41 million trainable parameters, is trained to predict the....

Did you experiment with other numbers of $T=1k$ noising steps? - I thought this number is usually just a few hundred so I wondered whether you might have done any ablation on this or whether there was a particular motivation for this comparatively high value.

We thank the reviewer for their detailed evaluation of our work. Our use of $T=1000$ is higher than other models typically adopt (RFDiffusion (Watson et al., 2023) for example, uses 200 steps). During initial development of our work, we experimented with various values of T but found that lower values tended to produce poorer losses on the held-out validation set. As such, we adopted a final value of $T=1000$.

I would recommend being more careful with any comparison to the natural folding process. You already kept it vague by saying “inspired by” but maybe consider to make even more clear that the natural folding process is very different from FoldingDiff to avoid any misunderstanding.

We agree with the recommendation regarding comparison to the natural folding process, and have further clarified throughout the manuscript text as follows:

In the third paragraph of our introduction:

.... Combining these ideas, our framework generates novel backbones by starting from a set of random angles that correspond to a random, unfolded state and iteratively denoising the underlying angles to arrive at a final backbone structure (Figure 1). *Although this angular denoising procedure does not directly capture any biophysical folding processes, it draws inspiration from how proteins twist and fold into their final structures; as such, we name our method FoldingDiff.*

In the first paragraph of our results section “Designing and training FoldingDiff”:

.... After training is complete, to generate new data points, the diffusion model starts from random noise and applies T steps of iterative denoising where the output of each prior denoising step is used to prepare the input for the next cycle of denoising, culminating in a “clean” sample (Algorithm 1, Figure 1d). *Importantly, this noising and denoising procedure does not approximate any biophysical processes of protein folding.*

In SOM you write “Even when considering longer structures up to 512 residues in length, the reconstructed structures still share a TMscore similarity much greater than 0.5 (graph not shown), which suggests that our method could scale up to larger structures.” - Given that most users would like to apply your method to proteins with more than 128 residues, it would be valuable to understand the trade-off between length and accumulated error beyond your training cut-off. As you already have the data for this, please add it to the figure. The same holds true for Fig. S3: given that your model uses relative positional encoding, FoldingDiff should be able to generalise to protein lengths not seen during training. To give readers better understanding to which extent length affects usefulness of FoldingDiff, please also expand Fig S3 up to e.g. L=512. This also gives a nice additional data point to which extent relative positional encoding actually extrapolates for this problem/setup.

We thank the reviewer for carefully evaluating all aspects of our work. We clarify that we do not make any claims surrounding FoldingDiff’s ability to scale beyond 128 residue proteins, and we have not run any analysis in this manner. Despite the fact that FoldingDiff uses a relative positional embedding, we expect that there would still be substantial challenges to scaling beyond 128 residues. Most notably, larger protein structures involve intricate interactions between secondary structure elements - interactions that are not as comprehensively captured given our training procedure’s cap at 128 residues. We have revised part of our Discussion to be more explicit about this limitation:

While we demonstrate promising results with our model, we note limitations to our method that motivate opportunities for future work. Although we show empirically that our angular representation can be trained to generate high-quality backbones with up to 128 residues, there is no guarantee that our framework scales effectively to longer backbone lengths or more complex applications,

We believe that this misunderstanding may come from our supplemental discussion that looks at converting CATH domains from coordinate to angular representations and back, which we do for proteins up to 512 residues in length. Notably, we do not train FoldingDiff beyond 128 residues; the aforementioned reconstruction error analysis is strictly analyzing potential errors in the representation itself, not in our model's ability to work effectively within that representation. We have updated the supplemental text to be more clear about this point and apologize for any misunderstandings.

... the reconstructed structures still share a TMscore similarity much greater than 0.5 (graph not shown), which suggests that our method could scale up to larger structures. Note, however, that our generative FoldingDiff model is currently only trained on structures up to 128 residues in length – this analysis focuses solely on the potential for errors introduced by our angular representation, and not FoldingDiff's ability to work effectively within this formulation or to design structures longer than 128 residues in length.

I think the manuscript would be more helpful for potential readers/users if the authors would provide a plot showing protein_length vs generation_time. This way, users could estimate how long generating e.g. 10 structures of a certain length takes.

We thank the reviewer for suggesting this helpful addition. We have added the following figure as the new Supplementary Figure S7 to our manuscript that describes the time required for sampling structures, as well as the accompanying caption:

FoldingDiff generation time. Time (y-axis) for FoldingDiff (blue) and RFDiffusion (latest version as of September 2023, orange) to unconditionally generate 100 structures of varying length (x-axis) on a machine with an Intel i9-9960X processor and a single NVIDIA 2080Ti GPU. Both methods leverage GPU acceleration for this benchmark, and are run using out-of-the-box settings. FoldingDiff's runtime grows approximately linearly with respect to size of the generated structure; generating 100 structures of 120 residues each takes about 80 seconds. RFDiffusion is slower and exhibits runtime scaling superlinear with respect to structure size; generating 100 structures of 120 residues each takes approximately 41 minutes.

We refer to this result in passing in our results section:

We unconditionally generated 10 backbone chains each for every length $l \in [50, 128]$ (see Methods, Figures 2a, S7), generating a total of 780 backbones.

The authors make the point that their method might capture protein dynamic and/or disorder. While I would understand if the authors were to leave this to future work, one way to easily quantify this would be to generate structures for the 117 proteins in this test set (<https://doi.org/10.1038/s41598-020-71716-1>) and correlate their residue-level structural fluctuation with the corresponding CheZOD-score. This would allow to put their performance directly into perspective of existing disorder predictors (and might give some support for the claim of capturing dynamics). But I consider the point on potential information leakage between train, test, val much more important, so I would understand if the authors leave this to future work (rather a suggestion to have a shortcut for assessment).

We appreciate the reviewer's thoughtful engagement and extensive suggestions for our work. We wish to clarify that FoldingDiff, in its current iteration, is designed to perform unconditional, unguided backbone generation, and as such has no knowledge of amino acid side chains or how protein sequence relates to structure. In our discussion, we do mention the possibility of extending FoldingDiff to perform sampling of structures conditioned on a particular amino acid sequence, but this is not within the scope of current work. However, as we pursue followup work, the dataset and evaluation approach mentioned by the reviewer will undoubtedly serve as a valuable resource.

We have updated the discussion text to be more clear that proteins with disordered regions are a direction for future work; please refer to the following updated text:

Further work that "guides" or biases the generative process towards backbones with desirable protein-protein interactions, structural domains, or functional traits will help realize the potential of proteins as therapeutic agents. Similarly, extending FoldingDiff to perform structure diffusion guided by amino acid sequence could enable new advances in protein fold prediction. Despite their lauded success, models like AlphaFold2 have been shown to perform poorly on proteins with highly dynamic folds, such as intrinsically disordered proteins [53, 54]. The conventional method for computationally modeling these systems involves extensive molecular dynamics simulations that are orders of magnitude slower than a typical deep learning model. A sequence-guided generative model of structure might bypass such expensive simulations and directly sample from the conditional distribution of conformations matching a given amino acid sequence. Indeed, some works have already begun to leverage FoldingDiff's ability to rapidly generate many backbones to model disordered protein structural ensembles [55].

REVIEWER COMMENTS

Reviewer #1 (Remarks to the Author):

I would like to thank the authors for the detailed revisions – they have addressed all of my points thoroughly. A few comments are listed below:

Atomic clashes. The increased number of clashes of structures generated by FoldingDiff is not surprising, especially when compared to a state-of-the-art model such as RFDiffusion. However, I believe this presents a more holistic view of FoldingDiff's performance. One minor comment is that it seems a bit odd to compare atomic clashes to RFDiffusion and not with ProtDiff and ProteinSGM that was used for scTM comparisons in the main text. It may be more appropriate to compare atomic clashes to ProtDiff, but I leave this for the authors to decide.

Lever arm effect. Although indeed the lever arm effect is clearly a drawback to the angular representation of FoldingDiff that is shown in the new analyses, I do agree this is now addressed and I commend the authors for their intellectual honesty here.

PyMOL visualization. The generated protein structures are definitely more visually appealing with the explicit annotation of secondary structure elements – I thank the authors for revising all of the figures to accommodate this.

UMAP projections. Figure S10c in the reviewer document contains 'generated' proteins marked as 'x', but this is missing in the figure of the main text.

Designability. It is a bit concerning to observe that the majority of structures considered 'designable' with an scTM > 0.5 are alpha-helical, although I do acknowledge that alpha helices are significantly more abundant than beta sheets. I would like to see a plot of scTM vs secondary structure to explicitly visualize this relationship.

I think these are all minor issues and I can recommend this paper for acceptance.

Reviewer #2 (Remarks to the Author):

I appreciate the authors' careful review and response to the Reviewer critiques. However, the author's new finding that > 50% of generated structures include atomic clashes (verses < 5% for the compared methods), in combination with their refusal to assess whether or not these backbone models can support sidechains, leads me to believe this work is not ready for publication in Nature Communications. I do not see sufficient evidence that these structures resemble real protein backbone structures.

The authors say about the clashes:

"However, these can be easily filtered out, or passed through structural refinement methods to resolve clashes"

This filtering process should be included if it is so easy, or alternatively, the structures should be refined as part of the method itself - the method does not need to stop at the diffusion model. In addition, as there are a number of methods for building sidechains onto backbones, the authors should show that some set of methods can build clash-free sidechains onto sequences predicted to have these backbone folds. With > 50% backbone clashes, I can't imagine there will be any structures that support sidechains without clashes. If this is the case, FoldDiff is not ready for publication, or should be published as a work-in-progress (which is not the goal of Nature Communications, as I understand it).

Reviewer #3 (Remarks to the Author):

Thanks for providing such an in-depth answer to my comments. Given that I understand what amount of work it would cause to redo the dataset as I had suggested, I consider all my concerns being addressed.

For the future, I would still suggest to consider evaluating the effect of a different test set split, i.e., a CATH-based one. Even if you argue that deep generative models struggle to generate true out-of-distribution (OOD) samples, I am unsure how to define the point at which OOD starts in our field.

Even proteins with different CATH topologies consist of smaller building blocks (e.g. stretches of k-mers forming similar/identical structural motifs) seen by the model. So the question is rather to which extent the model learnt to re-wire those building blocks in order to form structures not directly seen during training. Another aspect of such a test set would be that it might give a different angle on the correct early stopping checkpoint which your analysis showed to be important. So it might be interesting to see whether such a set might find a more optimal (potentially even earlier checkpoint) that is better at generalizing (maybe less prone towards generating helical structures).

In the following point-by-point response, we have reproduced the reviewer's original comment in **bold**, followed by normal text to indicate our response. Where appropriate, we reproduce passages from our main text inline to highlight updates; in these cases, the updated text is shown in blue.

Reviewer 1

I would like to thank the authors for the detailed revisions – they have addressed all of my points thoroughly. A few comments are listed below:

Atomic clashes. The increased number of clashes of structures generated by FoldingDiff is not surprising, especially when compared to a state-of-the-art model such as RFdiffusion. However, I believe this presents a more holistic view of FoldingDiff's performance. One minor comment is that it seems a bit odd to compare atomic clashes to RFdiffusion and not with ProtDiff and ProteinSGM that was used for scTM comparisons in the main text. It may be more appropriate to compare atomic clashes to ProtDiff, but I leave this for the authors to decide.

Lever arm effect. Although indeed the lever arm effect is clearly a drawback to the angular representation of FoldingDiff that is shown in the new analyses, I do agree this is now addressed and I commend the authors for their intellectual honesty here.

PyMOL visualization. The generated protein structures are definitely more visually appealing with the explicit annotation of secondary structure elements – I thank the authors for revising all of the figures to accommodate this.

We thank the reviewer for their support of our work and for the feedback above.

UMAP projections. Figure S10c in the reviewer document contains 'generated' proteins marked as 'x', but this is missing in the figure of the main text.

We apologize if there was an oversight in communicating this; we have double checked the manuscript, and the figure caption for the supplementary figure reads:

.... In panels (b-c), marker types denote whether each point corresponds to a CATH (circle) or FoldingDiff (cross) example.

The corresponding UMAP figure in the main text, Figure 3d-f, only includes generations from FoldingDiff (only the supplementary figure includes natural CATH structures for context). Thus, we do not describe the meaning of different plotting symbols in the main text. We hope this addresses the reviewer's point.

Designability. It is a bit concerning to observe that the majority of structures considered 'designable' with an scTM > 0.5 are alpha-helical, although I do acknowledge that alpha helixes

are significantly more abundant than beta sheets. I would like to see a plot of scTM vs secondary structure to explicitly visualize this relationship.

We agree with the reviewer that handling of beta sheets is an outstanding point for improvement in future work. As pointed out by the reviewer, we also suspect that alpha helices' relative abundance may play a role in this. We agree that a plot showing scTM as a function of secondary structure would be a strong addition, and as such have included this as a new Supplementary Figure S16, which is newly referenced on line 273 and has been reproduced below:

Figure S16. scTM designability scores broken down by secondary structure content. Relationship between training TM score similarity (x-axis) and scTM designability (y-axis) for FoldingDiff generated structures with at least one beta sheet (a) and with at least one alpha helix (b). Structures generated by FoldingDiff (Figure 4c, $n = 780$) are separated into two categories corresponding to (a) whether or not they contain at least one beta sheet (panel a, $n = 549$), and (b) whether not they contain at least one alpha helix (panel b, $n = 644$). These categories are non-exclusive – i.e., structures with at least one beta sheet may also contain alpha helices and vice versa.

Reviewer 2

I appreciate the authors' careful review and response to the Reviewer critiques. However, the author's new finding that > 50% of generated structures include atomic clashes (verses < 5% for the compared methods), in combination with their refusal to assess whether or not these backbone models can support sidechains, leads me to believe this work is not ready for publication in Nature Communications. I do not see sufficient evidence that these structures resemble real protein backbone structures.

The authors say about the clashes:

"However, these can be easily filtered out, or passed through structural refinement methods to resolve clashes"

This filtering process should be included if it is so easy, or alternatively, the structures should be refined as part of the method itself - the method does not need to stop at the diffusion model. In addition, as there are a number of methods for building sidechains onto backbones, the authors should show that some set of methods can build clash-free sidechains onto sequences predicted to have these backbone folds. With >50% backbone clashes, I can't imagine there will be any structures that support sidechains without clashes. If this is the case, FoldDiff is not ready for publication, or should be published as a work-in-progress (which is not the goal of Nature Communications, as I understand it).

We thank the reviewer for the feedback and agree with the suggestion that showing a set of methods for effectively building clash-free side chains onto our backbone structures would be a valuable addition to strengthen our work. As such, we have added new analysis to show that we can successfully use FASPR to add side chains to FoldingDiff's generated backbones according to amino acid sequences specified by ProteinMPNN, and then use PyRosetta to relax the full all-atom structure (including backbone atoms and side chain atoms). We show that these relaxed structures with side chain information are able to both (a) retain significant similarity to FoldingDiff's original backbone, and (b) eliminate nearly all atomic clashes.

Please see the following text, which has been added as a new section in the Supplementary Information entitled "Structural refinement".

FoldingDiff's raw generations can be passed through structural refinement methods to resolve clashes, taking inspiration from previous works [13]. To do this, we take each backbone generated by FoldingDiff and use ProteinMPNN to sample 8 different amino acid sequences predicted to fold to that backbone (this step and the sequences generated are shared with the designability analyses in the main text). We then use FASPR [71] to pack the side chains specified by each amino acid sequence onto the original backbone, yielding a total of 8 candidate all-atom structural models corresponding to the original backbone. We then apply PyRosetta's [72] FastRelax constrained relaxation protocol to each of these 8 candidate models and choose the relaxed structure with the highest TMscore to the original backbone as the final representative all-atom structure.

Comparing these representative all-atom structure models to original generations from FoldingDiff, we find that 455/780 generations have a TMscore ≥ 0.5 before and after attaching side chains with relaxation - i.e., most structures retained their overall fold. Furthermore, among these 455 structures, 433/455 (95.2%) are clash-free. Across all 780 relaxed structures, 733/780 (94.0%) are clash-free.

Together, these values indicate that side chain packing and subsequent relaxation effectively adds complete side chain orientations and resolves structural clashes while typically retaining the overall original generated structure. We also find that this relaxation TM score is highly correlated with scTM designability (Figure S5, Spearman's $\rho = 0.65$, $p = 8.61 \times 10^{-96}$), suggesting that side chain packing followed by structural relaxation may serve as a computationally faster and more robust (as it involves combining fewer deep learning models) method for evaluating quality of generated structures.

The referenced Figure S5 newly shows the correlation between this relaxation TM score and designability scTM score, and is reproduced below along with its caption:

Figure S5. Relationship between scTM designability and relaxation TM score. For each of the 780 structures generated and evaluated in the main text, a relaxation TM score was computed between the original backbone structure and the relaxed all-atom structure containing packed side chains. The amino acid sequences used for scTM analysis were considered, and rather than folding the sequences, FASPR [71] and PyRosetta [72] were used to pack the corresponding side chains directly onto FoldingDiff's generated backbone. After structural relaxation, the packed structure most similar to the original generation was used to calculate a relaxation TM score (y-axis) relative to the original backbone, and the relationship between the relaxation TM score (y-axis) and the scTM designability score (x-axis) was visualized.

Please also see the updated Table S1 below, which describes clash statistics, along with its updated caption.

Category	Total structures	Clash-free structures	Median clashes per structure
CATH	13963	13720 (98.3%)	0
RFDiffusion [18]	780	768 (98.5%)	0
FoldingDiff (w/o relax)	780	364 (46.7%)	2
FoldingDiff (relaxed)	455	433 (95.2%)	0

Table S1. Number of atomic clashes observed within CATH structures, FoldingDiff generations, and RFDiffusion generations of comparable length. The two rows for FoldingDiff correspond to structures that are directly generated without relaxation or refinement (“w/o relax”) and structures that have been structurally refined that also retain meaningful similarity to the original generation (“relaxed”).

All of these results are discussed within the main text, lines 103-106, which are reproduced below for reference:

To rule this out and confirm that our proposed representation can accurately describe longer protein structures, we convert a set of proteins of varying lengths from coordinate to angular representation and back, and find minimal differences between the original and reconstructed coordinates (Figure S2). We similarly investigate the potential for our angular formation to result in structures with atomic clashes, and find that although these clashes do appear, they can be easily remedied with common structural relaxation methods (see Supplementary Information; Figure S5, Table S1).

We believe that this analysis directly addresses the reviewer’s concerns that FoldingDiff’s backbones do not support side chains by showing that we can indeed do this.

Reviewer 3

Thanks for providing such an in-depth answer to my comments. Given that I understand what amount of work it would cause to redo the dataset as I had suggested, I consider all my concerns being addressed.

We thank the reviewer for their support of our work and for their feedback throughout this process.

For the future, I would still suggest to consider evaluating the effect of a different test set split, i.e., a CATH-based one. Even if you argue that deep generative models struggle to generate true out-of-distribution (OOD) samples, I am unsure how to define the point at which OOD starts in our field. Even proteins with different CATH topologies consist of smaller building blocks (e.g.

stretches of k-mers forming similar/identical structural motifs) seen by the model. So the question is rather to which extent the model learnt to re-wire those building blocks in order to form structures not directly seen during training. Another aspect of such a test set would be that it might give a different angle on the correct early stopping checkpoint which your analysis showed to be important. So it might be interesting to see whether such a set might find a more optimal (potentially even earlier checkpoint) that is better at generalizing (maybe less prone towards generating helical structures).

We agree that the process for selecting a test split that robustly captures generalization is an important, challenging task. We will be sure to keep the reviewer's valuable feedback in mind as we pursue followup work.

REVIEWERS' COMMENTS

Reviewer #2 (Remarks to the Author):

Thank you for responding to my feedback with the new relaxation step - this data greatly increasing my confidence in the method. No further comments.